# Learning Domain Invariant Representations in Goal-conditioned Block MDPs

**Beining Han**
IIIS, Tsinghua University
bouldinghan@gmail.com

**Chongyi Zheng**
Carnegie Mellon University
chongyiz@andrew.cmu.edu

**Harris Chan**   **Keiran Paster**   **Michael R. Zhang**   **Jimmy Ba**
University of Toronto & Vector Institute
{hchan, keirp, michael, jba}@cs.toronto.edu

## Abstract

Deep Reinforcement Learning (RL) is successful in solving many complex Markov Decision Processes (MDPs) problems. However, agents often face unanticipated environmental changes after deployment in the real world. These changes are often spurious and unrelated to the underlying problem, such as background shifts for visual input agents. Unfortunately, deep RL agents are usually sensitive to these changes and fail to act robustly against them. This resembles the problem of domain generalization in supervised learning. In this work, we study this problem for goal-conditioned RL agents. We propose a theoretical framework in the Block MDP setting that characterizes the generalizability of goal-conditioned policies to new environments. Under this framework, we develop a practical method *PA-SkewFit* that enhances domain generalization. The empirical evaluation shows that our goal-conditioned RL agent can perform well in various unseen test environments, improving by 50% over baselines.

## 1   Introduction

Deep Reinforcement Learning (RL) has achieved remarkable success in solving high-dimensional Markov Decision Processes (MDPs) problems, e.g., Alpha Zero Silver et al. [2017] for Go, DQN Mnih et al. [2015] for Atari games and SAC Haarnoja et al. [2018] for locomotion control. However, current RL algorithms requires massive amounts of trial and error to learn Silver et al. [2017], Mnih et al. [2015], Haarnoja et al. [2018]. They also tend to overfit to specific environments and often fail to generalize beyond the environment they were trained on Packer et al. [2018]. Unfortunately, this characteristic limits the applicability of RL algorithms for many real world applications. Deployed RL agents, e.g. robots in the field, will often face environment changes in their input such as different backgrounds, lighting conditions or object shapes Julian et al. [2020]. Many of these changes are often spurious and unrelated to the underlying task, e.g. control. However, RL agents trained without experiencing these changes are sensitive to the changes and often perform poorly in practice Julian et al. [2020], Zhang et al. [2020a,b].

In our work, we seek to tackle changing, diverse problems with goal-conditioned RL agents. Goal-conditioned Reinforcement Learning is a popular research topic as its formulation and method is practical for many robot learning problems Marcin et al. [2017], Eysenbach et al. [2020]. In goal-conditioned MDPs, the agent has to achieve a desired goal state $g$ which is sampled from a prior distribution. The agent should be able to achieve not only the training goals but also new test-time goals. Moreover, in practice, goal-conditioned RL agents often receive high-dimensional inputs for both observations and goals Paster et al. [2020], Péré et al. [2018]. Thus, it is important to ensure that

35th Conference on Neural Information Processing Systems (NeurIPS 2021).

the behaviour of goal-conditioned RL agents is invariant to any irrelevant environmental changes in the input at test time. Previous work Zhang et al. [2020a] tries to address these problems via model bisimulation metric Ferns et al. [2011]. These methods aim to acquire a minimal representation which is invariant to irrelevant environment factors. However, as goal-conditioned MDPs are a family of MDPs indexed by the goals, it is inefficient for these methods to acquire the model bisimulation representation for every possible goal, especially in high-dimensional continuous goal spaces (such as images).

In our work, we instead choose to optimize a surrogate objective to learn the invariant policy. Our main contributions are:

1. We formulate the Goal-conditioned Block MDPs (GBMDPs) to study domain generalization in the goal-conditioned reinforcement learning setting (Section 2), and propose a general theory characterizing how well a policy generalizes to unseen environments (Section 3.1).
2. We propose a theoretically-motivated algorithm based on optimizing a surrogate objective, *perfect alignment*, with *aligned data* (Section 3.2). We then describe a practical implementation based on Skew-Fit Pong et al. [2020] to achieve the objective (Section 3.3).
3. Empirically, our experiments for a sawyer arm robot simulation with visual observations and goals demonstrates that our proposed method achieves state-of-the-art performance compared to data augmentation and bisimulation baselines at generalizing to unseen test environments in goal-conditioned tasks (Section 4).

## 2   Problem Formulation

In this section, we formulate the domain invariant learning problem as solving Goal-conditioned Block MDPs (GBMDPs). This extends previous work on learning invariances Zhang et al. [2020a], Du et al. [2019] to the goal-conditioned setting Kaelbling [1993], Schaul et al. [2015], Marcin et al. [2017].

We consider a family of Goal-conditioned Block MDP environments $M^{\mathcal{E}} = \{(\mathcal{S}, \mathcal{A}, \mathcal{X}^e, \mathcal{T}^e, \mathcal{G}, \gamma) | e \in \mathcal{E}\}$ where $e$ stands for the environment index. Each environment consists of shared state space $\mathcal{S}$, shared action space $\mathcal{A}$, observation space $\mathcal{X}^e$, transition dynamic $\mathcal{T}^e$, shared goal space $\mathcal{G} \subset \mathcal{S}$ and the discount factor $\gamma$.

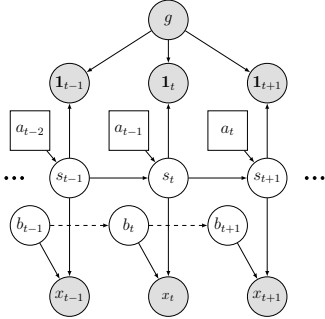

Moreover, we assume that $M^{\mathcal{E}}$ follows the generalized Block structure Zhang et al. [2020a]. The observation $x^e \in \mathcal{X}^e$ is determined by state $s \in \mathcal{S}$ and the environmental factor $b^e \in \mathcal{B}^e$, i.e., $x^e(s, b^e)$ (Figure 6(c)). For brevity, we use $x_t^e(s)$ to denote the observation for domain $e$ at state $s$ and step $t$. We may also omit $t$ as $x^e(s)$ if we do not emphasize on the step $t$ or the exact environmental factor $b_t^e$. The transition function is thus consists of state transition $p(s_{t+1}|s_t, a_t)$ (also $p(s_0)$), environmental factor transition $q^e(b_{t+1}^e|b_t^e)$. In our work, we assume the state transition is nearly deterministic, i.e., $\forall s, a$, entropy $\mathcal{H}(p(s_{t+1}|s_t, a_t)), \mathcal{H}(p(s_0)) \ll 1$, which is quite common in most RL benchmarks and applications Mnih et al. [2015], Greg et al. [2016], Pong et al. [2018]. Most

Figure 1: Graphical model for Goal-conditioned Block MDPs (GBMDPs) setting. The agent takes in the goal $g$ and observation $x_t$, which is produced by the domain invariant state $s_t$ and environmental state $b_t$, and acts with action $a_t$. Note that $b_t$ may have temporal dependence indicated by the dashed edge.

importantly, $\mathcal{X}^{\mathcal{E}} = \cup_{e \in \mathcal{E}} \mathcal{X}^e$ satisfies the disjoint property Du et al. [2019], i.e., each observation $x \in \mathcal{X}^{\mathcal{E}}$ uniquely determines its underlying state $s$. Thus, the observation space $\mathcal{X}^{\mathcal{E}}$ can be partitioned into disjoint blocks $\mathcal{X}(s), s \in \mathcal{S}$. This assumption prevents the partial observation problem.

The objective function in GBMDP is to learn a goal-conditioned policy $\pi(a|x^e, g)$ that maximizes the discounted state density function $J(\pi)$ Eysenbach et al. [2020] across all domains $e \in \mathcal{E}$. In our theoretical analysis, we do not assume the exact form of $g$ to the policy. One can regard $\pi(\cdot|x^e, g)$ as a group of RL policies indexed by the goal state $g$.

$$J(\pi) = \mathbb{E}_{e \sim \mathcal{E}, g \sim \mathcal{G}, \pi} \left[ (1 - \gamma) \sum_{t=0}^{\infty} \gamma^t p_\pi^e(s_t = g|g) \right] = \mathbb{E}_{e \sim \mathcal{E}}[J^e(\pi)] \tag{1}$$

$p_\pi^e(s_t = g|g)$ denotes the probability of achieving goal $g$ under policy $\pi(\cdot|x^e, g)$ at step $t$ in domain $e$. Besides, $e \sim \mathcal{E}$ and $g \sim \mathcal{G}$ refers to uniform samples from each set. As $p_\pi^e$ is defined over

state space, it may differs among environments since policy $\pi$ takes $x^e$ as input. Fortunately, in a GBMDP, there exist optimal policies $\pi_G(\cdot|x^e, g)$ which are invariant over all environments, i.e., $\pi_G(a|x^e(s), g) = \pi_G(a|x^{e'}(s), g), \forall a \in \mathcal{A}, s \in \mathcal{S}, e, e' \in \mathcal{E}$.

During training, the agent has access to training environments $\{e_i\}_{i=1}^N = \mathcal{E}_{\text{train}} \subset \mathcal{E}$ with their environment indices. However, we do not assume that $\mathcal{E}_{\text{train}}$ is i.i.d sampled from $\mathcal{E}$. Thus, we want the goal-conditioned RL agent to acquire the ability to neglect the spurious and unrelated environmental factor $b^e$ and capture the underlying invariant state information. This setup is adopted in many recent works such as in Zhang et al. [2020a] and in domain generalization Koh et al. [2020], Arjovsky et al. [2019] for supervised learning.

## 3 Method

In this section, we propose a novel learning algorithm to solve GBMDPs. First, we propose a general theory to characterize how well a policy $\pi$ generalizes to unseen test environments after training on $\mathcal{E}_{\text{train}}$. Then, we introduce *perfect alignment* as a surrogate objective for learning. This objective is supported by the generalization theory. Finally, we propose a practical method to acquire perfect alignment.

### 3.1 Domain Generalization Theory for GBMDP

In a seminal work, Ben-David et al. Ben-David et al. [2010] shows it is possible to bound the error of a classifier trained on a source domain on a target domain with a different data distribution. Follow-up work extends the theory to the domain generalization setting Sicilia et al. [2021], Albuquerque et al. [2019]. In GBMDP, we can also derive similar theory to characterize the generalization from training environments $\mathcal{E}_{\text{train}}$ to target test environment $t$. The theory relies on the Total Variation Distance $D_{\text{TV}}$ Wikipedia [2021] of two policies $\pi_1, \pi_2$ with input $(x^e, g)$, which is defined as follows.

$$D_{\text{TV}}(\pi_1(\cdot|x^e, g) \| \pi_2(\cdot|x^e, g)) = \sup_{A' \in \sigma(\mathcal{A})} |\pi_1(A'|x^e, g) - \pi_2(A'|x^e, g)|$$

In the following statements, we denote $\rho(x, g)$ as some joint distributions of goals and observations that $g \sim \mathcal{G}$ and $x$ is determined by $\rho(x|g)$. Additionally, we use $\rho_\pi^e(x^e|g)$ to denote the discounted occupancy measure of $x^e$ in environment $e$ under policy $\pi(\cdot|x^e, g)$ and refer $\rho_\pi^e(x^e)$ as the marginal distribution. Furthermore, we denote $\epsilon^{\rho(x,g)}(\pi_1 \| \pi_2)$ as the average $D_{\text{TV}}$ between $\pi_1$ and $\pi_2$, i.e., $\epsilon^{\rho(x,g)}(\pi_1 \| \pi_2) = \mathbb{E}_{\rho(x,g)}[D_{\text{TV}}(\pi_1(\cdot|x, g) \| \pi_2(\cdot|x, g))]$. This quantity is crucial in our theory as it can characterize the performance gap between two policies (see Appendix C).

Then, similar to the famous $\mathcal{H}\Delta\mathcal{H}$-divergence Ben-David et al. [2010], Sicilia et al. [2021] in domain adaptation theory, we define $\Pi\Delta\Pi$-divergence of two joint distributions $\rho(x, g)$ and $\rho(x, g)'$ in terms of the policy class $\Pi$:

$$d_{\Pi\Delta\Pi}(\rho(x, g), \rho(x, g)') = \sup_{\pi, \pi' \in \Pi} |\epsilon^{\rho(x,g)}(\pi \| \pi') - \epsilon^{\rho(x,g)'}(\pi \| \pi')|$$

On one hand, $d_{\Pi\Delta\Pi}$ is a distance metric which reflects the distance between two distributions w.r.t function class $\Pi$. On the other hand, if we fix these two distributions, it also reveals the quality of the function class $\Pi$, i.e., smaller $d_{\Pi\Delta\Pi}$ means more invariance to the distribution change. Finally, we state the following Proposition in which $\pi_G$ is some optimal and invariant policy.

**Proposition 1** (Informal). *For any $\pi \in \Pi$, we consider the occupancy measure $\{\rho_\pi^{e_i}(x^{e_i}, g)\}_{i=1}^N$ for training environments and $\rho_{\pi_G}^t(x^t, g)$ for the target environment. For simplicity, we use $\epsilon^{e_i}$ as the abbreviation of $\epsilon^{\rho_\pi^{e_i}(x^{e_i}, g)}$, $\epsilon^t$ as $\epsilon^{\rho_{\pi_G}^t(x^t, g)}$ and $\delta = \max_{e_i, e'_i \in \mathcal{E}_{train}} d_{\Pi\Delta\Pi}(\rho_\pi^{e_i}(x^{e_i}, g), \rho_\pi^{e'_i}(x^{e'_i}, g))$. Let*

$$\lambda = \frac{1}{N}\sum_{i=1}^N \epsilon^{e_i}(\pi^* \| \pi_G) + \epsilon^t(\pi^* \| \pi_G), \quad \pi^* = \arg\min_{\pi' \in \Pi} \sum_{i=1}^N \epsilon^{e_i}(\pi' \| \pi_G)$$

*Then, we have*

$$J^t(\pi_G) - J^t(\pi) \le \frac{1}{N}\sum_{i=1}^N \epsilon^{e_i}(\pi \| \pi_G) + \lambda + \delta + \min_{\rho(x,g) \in B} d_{\Pi\Delta\Pi}(\rho(x, g), \rho_{\pi_G}^t(x^t, g)) \quad (2)$$

*where $B$ is a characteristic set of joint distributions determined by $\mathcal{E}_{train}$ and policy class $\Pi$.*

The formal statement and the proof are shown in Appendix C.2. Generally speaking, the first term of the right hand side in Eq. (2) quantifies the performance of $\pi$ in the $N$ training environments. $\lambda$ quantifies the optimality of the policy class $\Pi$ over all environments. $\delta$ reflects how the policy class $\Pi$ can reflect the difference among $\{\rho_\pi^{e_i}(x^{e_i}, g), e_i \in \mathcal{E}_{\text{train}}\}$, which should be small if the policy class is invariant. The last term characterizes the distance between training environment and target environment and will be small if the training environments are diversely distributed.

Many works on domain generalization of supervised learning Ben-David et al. [2010], Liu et al. [2019], Sicilia et al. [2021], Albuquerque et al. [2019], Akuzawa et al. [2019] spend much effort in discussing the trade-offs among different terms similar to the ones in Eq. (2), e.g., minimizing $\delta$ may increase $\lambda$ Akuzawa et al. [2019], and in developing sophisticated techniques to optimize the bound, e.g. distribution matching Louizos et al. [2016], Li et al. [2018], Jin et al. [2020] or adversarial learning Liu et al. [2019].

Different from their perspectives, in GBMDPs, we propose a simple but effective criteria to minimize the bound. From now on, we only consider the policy class $\Pi = \Pi_\Phi = \{w(\Phi(x), g), \forall w\}$. Usually, $\Phi$ will be referred as an encoder which maps $x \in \mathcal{X}^\mathcal{E}$ to some latent representation $z = \Phi(x)$. We will also use the notation $z(s) = \Phi(x(s))$ if we do not emphasize on the specific environment.

**Definition 1** (**Perfect Alignment**). *An encoder is called a* perfect alignment *encoder $\Phi$ w.r.t environment set $E$ if $\forall e, e' \in E$ and $\forall s, s' \in \mathcal{S}$, $\Phi(x^e(s)) = \Phi(x^{e'}(s'))$ if and only if $s = s'$.*

As illustrated in Figure 5, an encoder is in perfect alignment if it maps two observations of the same underlying state $s$ to the same latent encoding $z(s)$ while also preventing meaningless embedding, i.e., mapping observations of different states to the same $z$. We believe perfect alignment plays an important role in domain generalization for goal-conditioned RL agents. Specifically, it can minimize the bound of Eq. (2) as follows.

**Proposition 2** (Informal). *If the encoder $\Phi$ is a perfect alignment over $\mathcal{E}_{train}$, then*

$$J^t(\pi_G) - J^t(\pi) \le \underbrace{\frac{1}{N} \sum_{i=1}^N \epsilon^{e_i}(\pi \parallel \pi_G)}_{(E)} + \underbrace{\epsilon^t(\pi^* \parallel \pi_G) + d_{\Pi_\Phi \Delta \Pi_\Phi}(\tilde{\rho}(x, g), \rho_{\pi_G}^t(x^t, g))}_{(t)} \quad (3)$$

*where $\tilde{\rho}(x, g)$ and $\pi^*$ are defined in Proposition 1 (also Appendix C).*

In Appendix C.3, we formally prove Proposition 2 when $\Phi$ is a $(\eta, \psi)$-perfect alignment, i.e., $\Phi$ is only near perfect alignment. The proof shows that the generalization error bound is minimized on the R.H.S of Eq. (3) when $\Phi$ asymptotically becomes an exact perfect alignment encoder. Therefore, in our following method, we aim to learn a perfect alignment encoder via aligned sampling (Section 3.2).

For the remaining terms in the R.H.S of Eq. (3), we find it hard to quantify them task agnostically, as similar difficulties also exist in the domain generalization theory of supervised learning Sicilia et al. [2021]. Fortunately, we can derive upper bounds for the remaining terms under certain assumptions and we observe that these upper bounds are significantly reduced via our method in the experiments (Section 4). The $(E)$ term represents how well the learnt policy $\pi$ approximates the optimal invariant policy on the training environments and is reduced to almost zero via RL (Table 1). For the $(t)$ term, we show that an upper bound of $(t)$ is proportion to the invariant quality of $\Phi$ on the target environment. Moreover, we find that learning a perfect alignment encoder over $\mathcal{E}_{\text{train}}$ empirically improves the invariant quality over other unseen environments $(t)$ (Figure 4). Thus, this $(t)$ term upperbound is reduced by learning perfect alignment. Please refer to Appendix C.4 for more details.

Based on the theory we derived in this subsection, we adopt perfect alignment as the heuristic to address GBMDPs in our work. In the following subsections, we propose a practical method to acquire a perfect alignment encoder over the training environments.

## 3.2 Learning Domain Invariant via Aligned Sampling

First, we discuss about the *if* condition on perfect alignment encoder $\Phi$, i.e., $\forall s, \Phi(x^e(s)) = \Phi(x^{e'}(s))$. The proposed method is based on *aligned sampling*. In contrast, most RL algorithms use observation-dependent sampling from the environment, e.g., $\epsilon$-greedy or Gaussian distribution policies Haarnoja et al. [2018], Fujimoto et al. [2018], Pong et al. [2020], Mnih et al. [2015]. However,

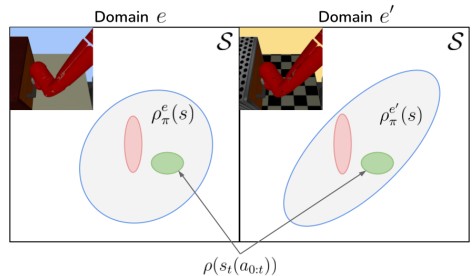
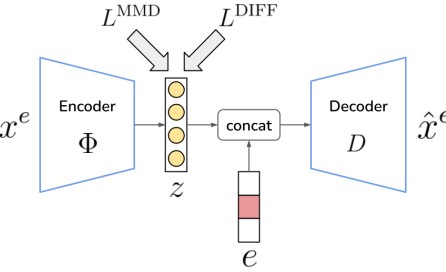

(a) Illustration of Aligned Sampling

(b) Overall structure

Figure 2: **(a)**: Illustration of Aligned Sampling. Square represents the whole state space $\mathcal{S}$, gray area represents the distribution $\rho_\pi^e(s)$ in two different environments. Small colored areas are the aligned state distribution generated by aligned sampling in Section 3.2. **(b)**: Overall VAE structure in our PA-SF. Encoder maps $x^e$ to the latent embedding $z$ and decoder $D$ reconstructs the observations with $z$ and index $e$. $L^{\text{MMD}}$ and $L^{\text{DIFF}}$ denote the two losses in Section 3.2.

with observation-dependent sampling, occupancy measures $\rho_\pi^e(s), \forall e \in \mathcal{E}_{\text{train}}$ will be different. Thus, simply aligning the latent representation of these observations will fail to produce a perfect alignment encoder $\Phi$.

Thus, we propose a novel strategy for data collection called *aligned sampling*. First, we randomly select a trajectory (e.g., from replay buffer etc.), denoted as $\{x_0^e, a_0, x_1^e, a_1, \ldots, x_T^e\}$ from environment $e$. The set of corresponding states along this trajectory are denoted as $\{s_t^e(a_{0:t})\}_{t=0}^T$. Second, we take the same action sequence $a_{0:T}$ in another domain $e'$ to get another trajectory $\{x_0^{e'}, a_0, x_1^{e'}, a_1, \ldots, x_T^{e'}\}$ (so as $\{s_t^{e'}(a_{0:t})\}_{t=0}^T$). We refer to the data collected by aligned sampling from all training environments as *aligned data*. These aligned observations $\{x_t^{e_i}(a_{0:t})\}, \forall e_i \in \mathcal{E}_{\text{train}}$ are stored in an aligned buffer $\mathcal{R}_{\text{align}}$ corresponding to the aligned action sequence $a_{0:t}$.

Under the definition of GBMDP, we have $\forall t \in [0:T], s \in \mathcal{S}, \rho(s_t^e(a_{0:t})) = \rho(s_t^{e'}(a_{0:t}))$, i.e., the same state distribution. Therefore, we can use MMD loss Gretton et al. [2008] to match distribution of $\Phi(x^e(s))$ for the aligned data. More specifically, in each iteration, we sample a mini-batch of $B$ aligned observations of every training environment $e_i \in \mathcal{E}_{\text{train}}$ from $\mathcal{R}_{\text{align}}$, i.e., $\mathcal{B}_{\text{align}} = \{x^{e_i}(s_t^{e_i}(a_{0:t}^b)), \forall e_i \in \mathcal{E}_{\text{train}}\}_{b=1}^B$. Then we use the following loss as a computationally efficient approximation of the MMD metric Zhao and Meng [2015], Louizos et al. [2016].

$$L^{\text{MMD}}(\Phi) = \mathbb{E}_{e,e' \sim \mathcal{E}_{\text{train}}, \mathcal{B}_{\text{align}} \sim \mathcal{R}_{\text{align}}} [\| \frac{1}{B} \sum_{b=1}^B \psi(\Phi(x^e(s_t^e(a_{0:t}^b)))) - \frac{1}{B} \sum_{b=1}^B \psi(\Phi(x^{e'}(s_t^{e'}(a_{0:t}^b)))) \|_2^2]$$

where $\psi$ is a random expansion function.

In Figure 2(a), we illustrate the intuition of the above approach. When the transition is nearly deterministic, the entropy for $\rho(s_t^e(a_{0:t}))$ is much smaller, i.e., $\mathcal{H}(\rho(s_t^e(a_{0:t}))) \ll \mathcal{H}(\rho_\pi^e(s_t))$. Thus, $\rho(s_t^e(a_{0:t}))$ can be regarded as small patches in $\mathcal{S}$. We use the MMD loss $L^{\text{MMD}}$ to match the latent representation $\{\Phi(x^e(s)), s \sim \rho(s_t^e(a_{0:t}))\}, \forall e \in \mathcal{E}_{\text{train}}$ together. As a consequence, we should achieve an encoder $\Phi$ that is more aligned. We discuss the theoretical property of $L^{\text{MMD}}$ in detail in Appendix C.5.

However, simply minimizing $L^{\text{MMD}}$ may violate the *only if* condition for perfect alignment. For example, a trivial solution for $L^{\text{MMD}} = 0$ is mapping all observations to some constant latent. To ensure that $\Phi(x^e(s)) = \Phi(x^{e'}(s'))$ *only if* $s = s'$, we additionally use the difference loss $L^{\text{DIFF}}$ as follows.

$$L^{\text{DIFF}}(\Phi) = -\mathbb{E}_{e \sim \mathcal{E}_{\text{train}}, x^e, \tilde{x}^e \in \mathcal{R}^e} \| \Phi(x^e) - \Phi(\tilde{x}^e) \|_2^2$$

where $\mathcal{R}^e$ refers to the replay buffer of environment $e$. Clearly, minimizing $L^{\text{DIFF}}$ encourages dispersed latent representations over all states $s \in \mathcal{S}$.

We refer to the combination $\alpha_{\text{MMD}} L^{\text{MMD}} + \alpha_{\text{DIFF}} L^{\text{DIFF}}$ as our perfect alignment loss $L^{\text{PA}}$. Note that $L^{\text{PA}}$ resembles contrastive learning Chen et al. [2020], Laskin et al. [2020a]. Namely, observations of aligned data from $\mathcal{R}_{\text{align}}$ are positive pairs while observations sampled randomly from a big replay

buffer are negative pairs. We match the latent embedding of positive pairs via the MMD loss while separating negative pairs via the difference loss. As discussed in Section 3.1, we believe this latent representation will improve generalization to unseen target environments.

### 3.3 Perfect Alignment for Skew-Fit

In Section 4, we will train goal-conditioned RL agents with perfect alignment encoder using the Skew-Fit algorithm Pong et al. [2020]. Skew-Fit is typically designed for visual-input agents which learn a goal-conditioned policy via purely self-supervised learning.

First, Skew-Fit trains a $\beta$-VAE with observations collected online to acquire a compact and meaningful latent representation for each state, i.e., $z(s)$ from the image observations $x(s)$. Then, Skew-Fit optimizes a SAC Haarnoja et al. [2018] agent in the goal-conditioned setting over the latent embedding of the image observation and goal, $\pi(a|z,g)$. The reward function is the negative of $l_2$ distance between the two latent representation $z(s)$ and $z(g)$, i.e., $r(s,g) = - \parallel z(s) - z(g) \parallel_2$. Furthermore, to improve sample efficiency, Skew-Fit proposes skewed sampling for goal-conditioned exploration.

In our algorithm, *Perfect Alignment for Skew-Fit* (PA-SF), the encoder $\Phi$ is optimized via both $\beta$-VAE losses as Pong et al. [2020], Nair et al. [2018] and $L^{\text{PA}}$ loss to ensure meaningful and perfectly aligned latent representation.

$$L(\Phi, D) = L^{\text{RECON}}(x^e, \hat{x}^e) + \beta D_{\text{KL}}(q_\Phi(z|x^e) \parallel p(z)) + \alpha_{\text{MMD}} L^{\text{MMD}} + \alpha_{\text{DIFF}} L^{\text{DIFF}} \quad (4)$$

In addition, we use both aligned sampling and observation-dependent sampling. Aligned sampling provides aligned data but hurts sample-efficiency while observation-dependent sampling is exploration-efficient but fails to ensure alignment. In practice, we find that collecting a small portion (15% of all data collected) of aligned data in $\mathcal{R}_{\text{align}}$ is enough for perfect alignment via $L^{\text{PA}}$.

Additionally, inspired by Louizos et al. [2016], we also change the $\beta$-VAE structure to what is shown in Figure 2(b), since in GBMDP data are collected from $N$ training environments and thus, the identity Gaussian distribution is no longer a proper fit for prior. The encoder $\Phi$ maps $x^e(s)$ to some latent representation $z(s)$ while the decoder $D$ takes both $z(s)$ and the environment index $e$ as input to reconstruct $\hat{x}^e(s)$. Note that by using both $L^{\text{PA}}$ and $L^{\text{RECON}}$, we require static environmental factors in $\mathcal{E}_{\text{train}}$ (unnecessary for testing environments) for a stable optimization. In future work, we will address the limit from $\beta$-VAE by training two latent representations simultaneously to stabilize the optimization for generality.

## 4 Experiments

In this section, we conduct experiments to evaluate our PA-SF algorithms. The experiments are based on multiworld Pong et al. [2018]. Our empirical analysis tries to answer the following questions: (1) How well does PA-SF perform in solving GBMDP problems? (2) How does each component proposed in Section 3 contribute to the performance?

### 4.1 Comparative Evaluation

In this subsection, we aim to answer the question (1) by comparing our proposed PA-SF method with vanilla Skew-Fit and several other baselines that attempt to acquire invariant policies for RL agents.

**Baselines** Current methods for obtaining robust policies can be characterized into two categories: (1) data augmentation and (2) model bisimulation.

1. **Data Augmentation**. Recent work Stone et al. [2021] tries to use data augmentation to prevent the RL agents from distractions. We implement the most widely accepted data augmentation methods RAD Laskin et al. [2020b] upon Skew-Fit (Skew-Fit + RAD) as a baseline. Note that our PA-SF method does not use any data augmentation and is parallel with this kind of techniques.

2. **Model Bisimulation** Ferns et al. [2011]. These methods utilize bisimulation metrics to learn a minimal but sufficient representation which will neglect irrelevant features of Block MDPs. We include MISA Zhang et al. [2020a] and DBC Zhang et al. [2020b] in our comparison as they are the most successful implementations for high-dimensional tasks. Moreover, in the goal-conditioned setting, we use an oracle state-goal distance $- \parallel s - g \parallel_2$ as rewards for these two algorithms in GBMDP. In contrast, our PA-SF method does not have such information.

Table 1: Evaluation of PA-SF and baselines on four control tasks. We report the mean and one standard deviation on each task (lower metric is better).

|  | Algorithm | Reach | Door | Push | Pickup |
|---|---|---|---|---|---|
|  |  | Hand distance (35K) | Angle difference (150K) | Puck distance (400K) | Object distance (280K) |
| Test Avg | Skew-Fit | $0.111 \pm 0.010$ | $0.194 \pm 0.018$ | $0.086 \pm 0.004$ | $0.037 \pm 0.006$ |
|  | Skew-Fit + RAD | $0.105 \pm 0.010$ | $0.162 \pm 0.030$ | $0.082 \pm 0.008$ | $0.040 \pm 0.004$ |
|  | MISA | $0.239 \pm 0.0142$ | $0.255 \pm 0.027$ | $0.099 \pm 0.006$ | $0.043 \pm 0.004$ |
|  | DBC | $0.185 \pm 0.037$ | $0.320 \pm 0.033$ | $0.095 \pm 0.006$ | $0.045 \pm 0.002$ |
|  | PA-SF(**Ours**) | $\mathbf{0.076} \pm 0.005$ | $\mathbf{0.106} \pm 0.015$ | $\mathbf{0.069} \pm 0.005$ | $\mathbf{0.028} \pm 0.004$ |
| Train Avg | PA-SF(**Ours**) | $0.067 \pm 0.005$ | $0.058 \pm 0.074$ | $0.060 \pm 0.005$ | $0.020 \pm 0.008$ |
|  | Oracle Skew-Fit | $0.055 \pm 0.010$ | $0.057 \pm 0.012$ | $0.054 \pm 0.006$ | $0.020 \pm 0.006$ |

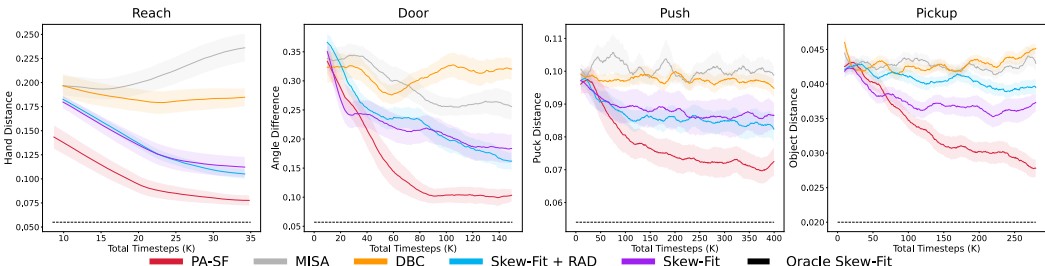

Figure 3: Learning curve of all algorithms on average across test environments for each task. All curves show the mean and one standard deviation (a half for Pickup to show clearly) of 7 seeds.

**Environments** We evaluate PA-SF and all baselines on a set of GBMDP tasks based on multiworld benchmark Pong et al. [2018], which is widely used to evaluate the performance of visual input goal-conditioned algorithms. We use the following four basic tasks Nair et al. [2018], Pong et al. [2020]: *Reach*, *Door*, *Pickup* and *Push*. In GBMDP, we create different environments with various backgrounds, desk surfaces, and object appearances. During testing, we also create environments with unseen video backgrounds to mimic environmental factor transitions $q^e(b_{t+1}^e|b_t^e)$. This makes policy generalization more challenging. Please refer to Appendix E for a full description of our experiment setup and implementation details of the baselines and our algorithm.

**Results** In Table 1, we show the final average performance of each algorithm on *unseen* test environments $\mathcal{E}_{\text{test}}$. The corresponding learning curves are shown in Figure 3. This metric shows the generalizability of each RL agent. All these agents are allowed to collect data from $\mathcal{E}_{\text{train}}$ ($N = 3$) with static environmental factors. Our PA-SF achieves SOTA performance on all tasks. On testing environments, we achieve a relative reduction around 40% to 65% of the corresponding metrics over vanilla Skew-Fit w.r.t the optimal metric possible (Oracle Skew-Fit). Oracle Skew-Fit refers to the performance of a Skew-Fit algorithm trained directly on the single environment (and *not* simultaneously on all $\mathcal{E}_{\text{train}}$).

Other invariant policy learning methods perform sluggishly on all tasks. For DBC and MISA, we hypothesize that they struggle for goal-conditioned problems since the model bisimulation metric is defined for a single MDP. In GBMDPs, this means acquiring a set of encoders $\Phi_g$ that achieves model bisimulation for every possible $g$ and is thus inefficient for learning. By design, our method is not susceptible to this issue as the perfect alignment is a universal invariant representation for all goals. Data augmentation via RAD provides marginal improvement over the vanilla Skew-Fit. Nevertheless, we believe developing adequate data augmentation techniques for GBMDPs is an important research problem and is orthogonal with our method.

Additionally, we also show the performance of PA-SF on the training environments in Table 1. PA-SF is still comparable and as sample-efficient as Skew-Fit in the training environments. This supports the claim that the $(E)$ term in the R.H.S of Eq. (3) is reduced to almost zero via RL training in practice.

## 4.2 Design Evaluation

In this subsection, we conduct comprehensive analysis on the design of PA-SF to interpret how well it carries out the theoretical framework discussed in Section 3.1 and Section 3.2.

To begin with, we show the learning curves in Figure 4 of different ablations of PA-SF in the *Door* environment during both training and testing. To understand the roles of $L^{\text{DIFF}}$ and $L^{\text{MMD}}$, PA-SF (w/o D) excludes $L^{\text{DIFF}}$ and PA-SF (w/o MD) excludes both losses[1]. Noticing that PA-SF (w/o MD) is equivalent to the Skew-Fit algorithm with our proposed VAE structure (Figure 2(b)). We also add PA-SF (w/o AS) which excludes aligned sampling.

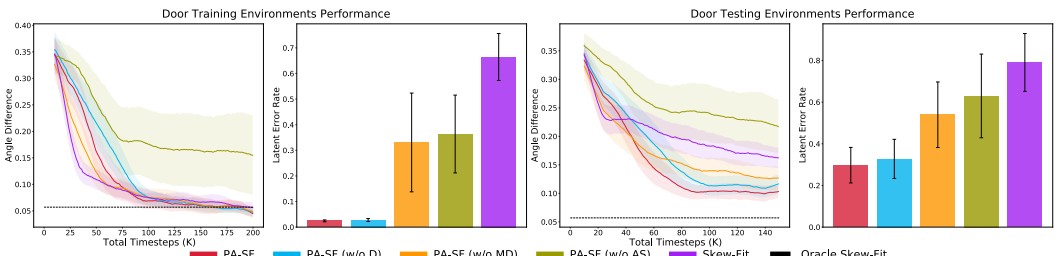

Figure 4: Ablation of PA-SF and visualization of the latent representation via LER metric. All curves represent the mean and one standard deviation across 7 seeds.

Additionally, we also quantify the quality of the latent representation $\Phi(x^e(s))$ in Figure 4 via the metric *Latent Error Rate* (LER). LER is defined as the average over environment set $E \in \{\mathcal{E}_{\text{train}}, \mathcal{E}_{\text{test}}\}$ as follows:

$$\text{Err}(\Phi) = \mathbb{E}_{e \sim E, s \sim \mathcal{S}} \left[ \frac{\| \Phi(x^e(s)) - \Phi(x^{e_0}(s)) \|_2}{\| \Phi(x^e(s)) \|_2} \right]$$

In general, the smaller $\text{Err}(\Phi)$ is, the closer the encoder $\Phi$ is to perfect alignment over the environments $E$. We first focus on the discussion about training performance.

1. $\Phi$ achieves the *if* condition of perfect alignment over $\mathcal{E}_{\text{train}}$ via $L^{\text{MMD}}$ as the LER value of PA-SF and PA-SF (w/o D) is almost 0. While without MMD loss, PA-SF (w/o MD) and Skew-Fit struggle with large LER value despite achieving good training performance. Furthermore, the comparison between PA-SF and PA-SF (w/o AS) demonstrates the importance of using aligned data in the MMD loss (Otherwise, the matching is inherently erroneous).

2. The *only if* condition, i.e., $\Phi(x^e(s)) = \Phi(x^{e'}(s'))$ only if $s = s'$, is also achieved empirically by visualizing the reconstruction of the VAE (Figure 9 in Appendix D) and we believe this is satisfied by both the difference loss and the reconstruction loss. Under the *only if* condition, the SAC Haarnoja et al. [2018] trained on the latent space achieves the optimal performance. In contrast, PA-SF (w/o AS) fails to learn well on the training environments as its latent representation is mixed over different states.

Second, we focus on the generalization performance on target domains $t$, i.e., term $(t)$ in Eq. (3). We observe the following:

1. As shown by the learning curve of test environments, the target domain performance of different ablations match that of the LER metric: SkewFit, PA-SF (w/o AS) > PA-SF (w/o MD) > PA-SF (w/o D) > PA-SF. During training, these ablations have almost the same performance, except PA-SF (w/o AS). This indicates that the increased test performance indeed comes from the improved representation quality of the encoder $\Phi$, i.e., more aligned. This supports our claim at the end of Section 3.1 and the upper bound analysis on the $(t)$ term in Appendix C.4, that the increased invariant property of $\Phi$ produces better domain generalization performance.

2. In test environment ablations, the LER is reduced significantly on methods with $L^{\text{MMD}}$. This supports our claim that a perfect alignment encoder on training environments also improves the encoder's invariant property on unseen environments. In addition, by encouraging dispersed latent representation, the difference loss $L^{\text{DIFF}}$ also plays a role in reducing LER during testing. This supports the necessity of both losses for generalization.

---

[1] A single $L^{\text{DIFF}}$ is not useful here.

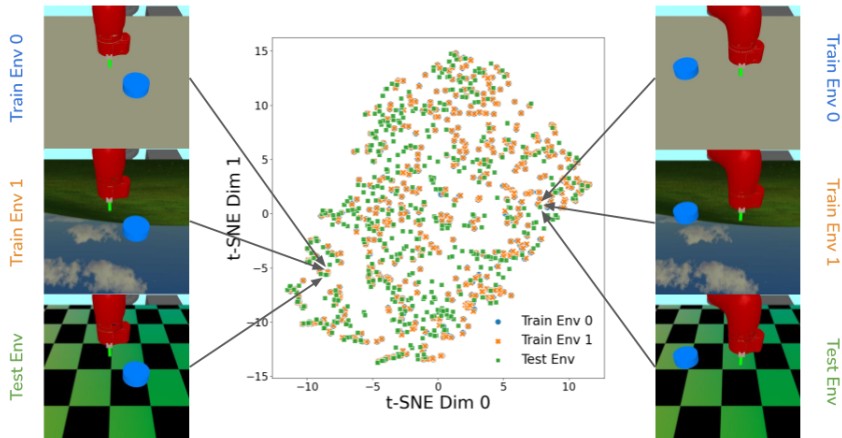

Figure 5: t-SNE visualization of the latent space $\Phi(x^e)$ trained with PA-SF for three environments: 2 training and 1 testing of *Push* as well as instances visualization.

We observe the similar results in other tasks as well (Appendix D). Here, we also visualize the latent space by t-SNE plot to illustrate the perfect alignment on task *Push*. Dots in training environments are matched perfectly and the corresponding test environment dot is approximately near as expected.

## 5 Related Work

**Goal-conditioned RL:** Goal-conditioned RL Kaelbling [1993], Schaul et al. [2015] removes the need for complicated reward shaping by only rewarding agents for reaching a desired set of goal states. RIG Nair et al. [2018] is the seminal work for visual-input, reward-free learning in goal-conditioned MDPs. Skew-Fit Pong et al. [2020] improves over RIG Nair et al. [2018] in training efficiency by ensuring the behavioural goals used to explore are diverse and have wide state coverage. However, Skew-Fit has its own limitation in understanding the semantic meaning of the goal-conditioned task. To acquire more meaningful goal's and observation's latent representation, several approaches apply inductive biases or seek human feedback. ROLL Wang et al. [2020] applies object extraction methods under strong assumptions, while WSC Lee et al. [2020] uses weak binary labeled data as the reward function. Others explore the same goal-conditioned RL problem via hindsight experience replay Marcin et al. [2017], Ren et al. [2019], Ghosh et al. [2019], unsupervised reward learning Péré et al. [2018], inverse dynamics models Paster et al. [2020], C-learning Eysenbach et al. [2020], goal generation Florensa et al. [2018], Nair and Finn [2019], Pitis et al. [2020], goal-conditioned forward models Nair et al. [2020], and hierarchical RL Li et al., Nachum et al. [2018], Zhang et al. [2020c], Hou et al. [2020]. Our study focus on learning goal-conditioned policies that is invariant of spurious environmental factors. We aim to learn a policy that can generalize to visual goals in unseen test environments.

**Learning Invariants in RL:** Robustness to domain shifts is crucial for real-world applications of RL. Zhang et al. [2020a,b], Gelada et al. [2019] implement the model-bisimulation framework Ferns et al. [2011] to acquire a minimal but sufficient representation for solving the MDP problem. However, model-bisimulation for high-dimension problems typically requires domain-invariant and dense rewards. These assumptions do not hold in GBMDPs. Contrastive Metric Embeddings (CME) Agarwal et al. [2021] instead uses $\pi^*$-bisimulation metric but it also requires extra information of the optimal policy. Another line of work tries to address these issues via self-supervised learning. Stone et al. [2021] tests multiple data augmentation methods including RAD Laskin et al. [2020b] and DrQ Kostrikov et al. [2020] to boost the robustness of the representation as well as the policy. Our work can also apply data augmentation in practice. However, we find that RAD is not very helpful in the Skew-Fit framework. Additionally, Hansen et al. [2020], Bodnar et al. [2020] use self-supervised correction during real-world adaptation like sim2real transfer but these methods are incompatible for domain generalization.

# 6 Conclusion

In this paper, we study the problem of learning invariant policies in Goal-conditioned RL agents. The problem is formulated as a GBMDP, which is an extension of Goal-conditioned MDPs and Block MDPs where we want the agent's policy to generalize to unseen test environments after training on several training environments.

As supported by the generalization bound for GBMDP, we propose a simple but effective heuristic, i.e., perfect alignment which we can minimize the bound asymptotically and benefit the generalization. To learn a perfect alignment encoder, we propose a practical method based on aligned sampling. The method resembles contrastive learning: matching latent representation of aligned data via MMD loss and dispersing the whole latent representations via the DIFF loss. Finally, we propose a practical implementation Perfect Alignment for Skew-Fit (PA-SF) by adding the perfect alignment loss to Skew-Fit and changing the VAE structure to handle GBMDPs.

The empirical evaluation shows that our method is the SOTA algorithm and achieves a remarkable increase in test environments' performance over other methods. We also compare our algorithm with several ablations and analyze the representation quantitatively. The results support our claims in the theoretical analysis that perfect alignment criteria is effective and that we can effectively optimize the criteria with our proposed method. We believe the perfect alignment criteria will enable applications in diverse problem settings and offers interesting directions for future work, such as extensions to other goal-conditioned learning frameworks Eysenbach et al. [2020], Paster et al. [2020].

## Acknowledgements

We are grateful for the feedback from anonymous reviewers. Resources used in preparing this research were provided, in part, by the Province of Ontario, the Government of Canada through CIFAR, and companies sponsoring the Vector Institute <http://www.vectorinstitute.ai/partners>.

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
