# A Notation

Table 2: Description for symbols

| Symbol | Description |
| --- | --- |
| $M^{\mathcal{E}}$ | A family of Goal-conditioned Block MDPs |
| $e$ | Environment index |
| $\mathcal{S}$ | Shared state space among environments $e \in \mathcal{E}$ |
| $\mathcal{A}$ | Shared action space among environments $e \in \mathcal{E}$ |
| $\mathcal{X}^e$ | Specific observation space for environment $e$ |
| $\mathcal{T}^e$ | Specific transition dynamic for environment $e$ |
| $\mathcal{G}$ | Shared goal space among environments $e \in \mathcal{E}$ |
| $\gamma$ | Shared discount factor among environments $e \in \mathcal{E}$ |
| $b^e$ | Environmental factor for environment $e$ (e.g. background) |
| $\mathcal{B}^e$ | Specific environmental factor space for environment $e$ (i.e. video backgrounds are allowed) |
| $x_t^e = x^e(s_t, b_t^e)$ | Observation determined by state $s$ and environmental factor $b^e$ for environment $e$ at timestep $t$ |
| $p(s_{t+1}|s_t, a_t)$ | State transition shared among environments |
| $q^e(b_{t+1}^e|b_t^e)$ | Environmental factor transition for environment $e$ |
| $\mathcal{X}^{\mathcal{E}} = \cup_{e \in \mathcal{E}} \mathcal{X}^e$ | Joint set of observation spaces |
| $\pi(a|x^e, g)$ | Goal-conditioned policy shared among environments |
| $J(\pi)$ | Objective function for policy $\pi$ |
| $J^e(\pi)$ | Objective function for policy $\pi$ in environment $e$ |
| $p_\pi^e(s_t = g|g)$ | Probability of achieving goal $g$ under policy $\pi(\cdot|x^e, g)$ at timestep $t$ in environment $e$ |
| $\pi_G(\cdot|x^e, g)$ | Optimal policies which are invariant over all environments |
| $\{e_i\}_{i=1}^N = \mathcal{E}_{\text{train}}$ | Training environments |
| $\rho(x, g) = \rho(g)\rho(x|g)$ | Joint distributions of goals and observations |
| $\rho_\pi^e(x^e|g)$ | Occupancy measure of $x^e$ in environment $e$ under policy $\pi(\cdot|x^e, g)$ |
| $\rho_\pi^e(x^e)$ | Marginal distribution of $\rho_\pi^e(x^e, g)$ over goals |
| $\epsilon^{\rho(x,g)}(\pi_1 \parallel \pi_2)$ | Averaged Total Variation between policy $\pi_1$ and $\pi_2$ |
| $\Pi$ | Policy class (i.e. space for all possible policies) |
| $d_{\Pi \Delta \Pi}(\rho(x, g), \rho(x, g)')$ | $\Pi \Delta \Pi$-divergence of two joint distributions $\rho(x, g)$ and $\rho(x, g)'$ in terms of the policy class $\Pi$ |
| $\epsilon^{e_i}(\pi_1 \parallel \pi_2) = \epsilon^{\rho_\pi^{e_i}(x^{e_i}, g)}(\pi_1 \parallel \pi_2)$ | Total Variation between policy $\pi_1$ and $\pi_2$ averaged over joint occupancy measure under policy $\pi$ in training environment $e_i$ |
| $\epsilon^t(\pi_1 \parallel \pi_2) = \epsilon^{\rho_{\pi_G}^t(x^t, g)}(\pi_1 \parallel \pi_2)$ | Total Variation between policy $\pi_1$ and $\pi_2$ averaged over joint occupancy measure under policy $\pi_G$ in testing environment $t$ |
| $\pi^*$ | The closest policy for training environments in policy class $\Pi$ w.r.t.optimal invariant policy $\pi_G$ measured by averaged Total Variation |

Continued on next page

| Symbol | Description |
|---|---|
| $\delta$ | Maximum $\Pi\Delta\Pi$-divergence between occupancy measure for two different training environments under given policy $\pi$ |
| $\lambda$ | Performance of $\pi^*$ in both training and testing environments in terms of average TV distance |
| $B$ | Characteristic set of joint distributions determined by $\mathcal{E}_{\text{train}}$ and policy class $\Pi$ |
| $\Phi$ | Observation encoder |
| $z^e(s) = \Phi(x^e(s))$ | Latent representation of observation $x^e$ with state $s$ |
| $\Pi_\Phi = \{w \circ (\Phi(x), g)\}, \forall w\}$ | Policy class induced by encoder $\Phi$ with any function $w$ |
| $\tilde{\rho}(x, g)$ | The closest occupancy measure in characteristic set $B$ w.r.t. occupancy measure in testing environment under $\Pi\Delta\Pi$-divergence |
| $\{s_t^e(a_{0:t})\}_{t=0}^T$ | Set of states along trajectory $\{x_0^e, a_0, x_1^e, a_1, \ldots, x_T^e\}$ |
| $\{x_t^{e_i}(a_{0:t})\}$ | Aligned observations in environment $e_i$ for action sequence $\{a_0, \ldots, a_{t-1}\}$ (`numpy` style indexing) |
| $\mathcal{R}_{\text{align}}$ | Replay buffer for aligned transitions |
| $\mathcal{B}_{\text{align}} = \{x^{e_i}(s_t^{e_i}(a_{0:t}^b)), \forall e_i \in \mathcal{E}_{\text{train}}\}_{b=1}^B$ | Batch of aligned observations from all the training environments |
| $L^{\text{MMD}}(\Phi)$ | MMD loss for encoder $\Phi$ |
| $\psi(z)$ | Random expansion function for latent representation $z$ |
| $L^{\text{DIFF}}(\Phi)$ | Difference loss for encoder $\Phi$ |
| $\mathcal{R}^e$ | Replay buffer for transitions from environment $e$ |
| $L^{\text{PA}}$ | Perfect alignement loss |
| $L^{\text{RECON}}$ | Reconstruction loss |
| $\beta$ | KL divergence coefficient |
| $\alpha_{\text{MMD}}$ | MMD loss coefficient |
| $\alpha_{\text{DIFF}}$ | Difference loss coefficient |
| $\text{Err}(\Phi)$ | Latent error rate for encoder $\Phi$ |

# B  Algorithm

The main difference between the PA-SF and Skew-Fit are (i) separate replay buffer for each training environments $\mathcal{R} = \{\mathcal{R}^e, e \in \mathcal{E}_{\text{train}}\}$, (ii) an additional aligned buffer for the aligned data $\mathcal{R}_{\text{align}} = \{\mathcal{R}^e_{\text{align}}, e \in \mathcal{E}_{\text{train}}\}$ and a corresponding aligned sampling procedure, (iii) VAE training uses Eq. (4) with mini-batches from both replay buffer and aligned buffer. The overall algorithm is described in Algorithm 1 and implementation details are listed in Appendix E.

---

**Algorithm 1** Perfect Alignment for Skew-Fit (PA-SF).

---

**Require:** $\beta$-VAE decoder, encoder $q_\phi$, goal-conditioned policy $\pi_\theta$, goal-conditioned value function $Q_w$, skew parameter $\alpha$, VAE training schedule, training environments $\mathcal{E}_{\text{train}}$, replay buffer $\mathcal{R} = \{\mathcal{R}^e, e \in \mathcal{E}_{\text{train}}\}$, aligned buffer $\mathcal{R}_{\text{align}} = \{\mathcal{R}^e_{\text{align}}, e \in \mathcal{E}_{\text{train}}\}$, coefficients in Eq. (4).

1: **for** $m = 0, \ldots, M - 1$ episodes **do**
2:     **for** $e = 0, \ldots, N - 1$ training environments **do**             ▷ Exploration Rollout
3:         Sample goal observation $x^e(g) \sim p^{e,m}_{\text{skewed}}$ and encode as $z^e(g) = q_\phi(x^e(g))$.
4:         Sample initial observation $x^e_0$ from the environment $e$.
5:         **for** $t = 0, \ldots, H - 1$ steps **do**
6:             Get action $a_t \sim \pi_\theta(q_\phi(x^e_t), g)$.
7:             Get next state $x^e_{t+1} \sim p(\cdot \mid x^e_t, a_t)$.
8:             Store $(x^e_t, a_t, x^e_{t+1}, x^e(g))$ into replay buffer $\mathcal{R}^e$.
9:         **end for**
10:     **end for**
11:     Sample action sequences $\{a_{0:T}\}$ by executing the policy
        in a random training environment.                      ▷ Aligned Sampling
12:     **for** $e = 0, \ldots, N - 1$ training environments **do**
13:         Sample initial state $x^e_0(s^e_0)$ from the environment $e$.
14:         Rollout action sequence $\{a_{0:T}\}$ to get $\{x^e_t(s^e_t(a_{0:t}))\}^T_{t=0}$.
15:     **end for**
16:     Store $\{x^e_t(s^e_t(a_{0:t}))\}^T_{t=0}$ in aligned buffer $\mathcal{R}^e_{\text{align}}$ indexed by $a_{0:t}$ for $e \in \mathcal{E}_{\text{train}}$.
17:     **for** $i = 0, \ldots, I - 1$ training iterations **do**           ▷ Policy Gradient
18:         Sample transition $(x^e_{t'}, a_{t'}, x^e_{t'+1}, z^e(g)) \sim \mathcal{R}^e$ for all $e \in \mathcal{E}_{\text{train}}$.
19:         Encode $z^e_{t'} = q_\phi(x^e_{t'}), z^e_{t'+1} = q_\phi(x^e_{t'+1})$.
20:         (Probability 0.5) replace $z^e(g)$ with $q_\phi(x'(g))$ where $x'(g) \sim p^{e,m}_{\text{skewed}}$.
21:         Compute new reward $r = -\|z^e_{t'+1} - z^e(g)\|_2$.
22:         Update $\pi_\theta$ and $Q_w$ via SAC on $(z^e_{t'}, a_{t'}, z^e_{t'+1}, z^e(g), r)$.
23:     **end for**
24:     **for** $t = 0, ..., H - 1$ steps **do**               ▷ Hindsight Relabeling
25:         **for** $j = 0, ..., J - 1$ steps **do**
26:             Sample future state $x^e_{h_j}, t < h_j \leq H - 1$ for all $e \in \mathcal{E}_{\text{train}}$.
27:             Store $\left(x^e_t, a_t, x^e_{t+1}, q_\phi\left(x^e_{h_j}\right)\right)$ into $\mathcal{R}^e$.
28:         **end for**
29:     **end for**
30:     Construct skewed replay buffer distribution $p^{e,m+1}_{\text{skewed}}$ using data
        from $\mathcal{R}^e$ for all $e \in \mathcal{E}_{\text{train}}$.                   ▷ Skewing Replay Buffers
31:     Fine-tune $\beta$-VAE on $\{x^e\}^B_{b=1} \sim p^{e,m+1}_{\text{skewed}}$ and $\{x^e(s^e_t(a^b_{0:t}))\}^B_{b=1} \sim \mathcal{R}^e_{\text{aligned}}$
        for all $e \in \mathcal{E}_{\text{train}}$ according to the VAE training schedule and via Eq. (4).      ▷ VAE Training
32: **end for**

---

## C  Proofs and Discussions

In this section, we provide detailed proofs and statements omitted in the main text. In addition, we also discuss the assumptions we make in the analysis in detail.

### C.1  Illustration of Different MDP Problems

Here, we illustrate different graphical models of related MDPs including Block MDPs (Figure 6(a)), Goal-conditioned MDPs (Figure 6(b)), and ours Goal-conditioned Block MDPs (GBMDP) (Figure 6(c)). We use the indicator funtion in the Goal-conditioned and GBMDP settings to emphasize that the reward is sparse. In practice, the goal $g$ may only be indirectly observed as $x^e(g)$, such as future state in pixel space for a particular domain.

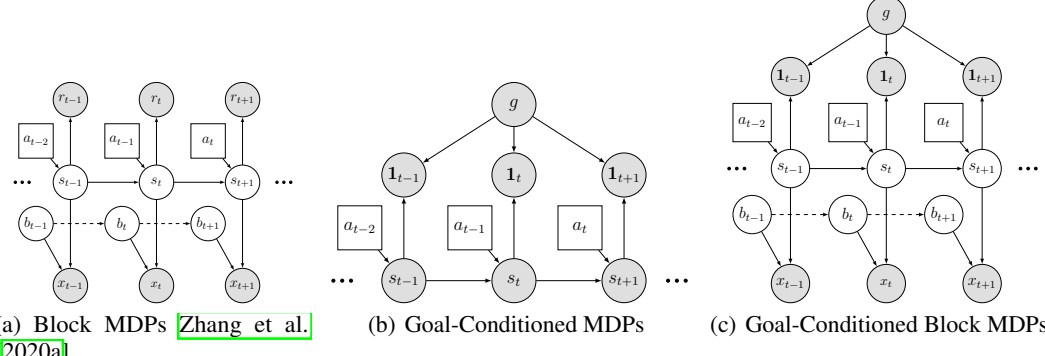

(a) Block MDPs Zhang et al. [2020a]  (b) Goal-Conditioned MDPs  (c) Goal-Conditioned Block MDPs

Figure 6: Comparison of graphical models of **(a)** a Block MDP Du et al. [2019], Zhang et al. [2020a], **(b)** a Goal-Conditioned MDP Kaelbling [1993], Schaul et al. [2015], Marcin et al. [2017], and **(c)** our proposed Goal-Conditioned Block MDP. The agent takes in the goal $g$ and observation $x_t$, which is produced by the domain invariant state $s_t$ and environmental state $b_t$, and acts with action $a_t$. $\mathbf{1}_t$ denotes the indicator function on whether the inputs are the same state. Note that $b_t$ may have temporal dependence indicated by the dashed edge.

### C.2  Proof of Proposition 1

Recall that Proposition 1 bounds the generalization performance by 4 terms: (1) average training environments' performance, (2) optimality of the policy class, (3) $d_{\Pi \Delta \Pi}$ over all training environments, and (4) the discrepancy measure between training environments and the target environment.

We begin our analysis by proving the following two Lemmas. For simplicity, we denote $p_\pi^{\Delta,e}(s|g)$ as the *discounted state density* as follows.

$$p_\pi^{\Delta,e}(s|g) = (1-\gamma) \sum_{t=0}^{\infty} \gamma^t p_\pi^e(s_t = s|g)$$

where $p_\pi^e(s_t = s|g)$ is the probability of state $s$ under goal-conditioned policy $\pi(\cdot|x^e, g)$ at step $t$ in domain $e$ (marginalized over the initial state $s_0 \sim p(s_0)$, previous actions $a_i \sim \pi(a_i|x_i^e, g), i = 0, \ldots, t-1$, and previous states $s_i \sim p(s_i|s_{i-1}, a_{i-1}), i = 0, \ldots, t-1$).

**Lemma 1.** $\forall e \in \mathcal{E}_{all}$, let $\rho_\pi^e(x^e, g)$ denote joint distributions of $g \sim \mathcal{G}$ and $x^e$ under policy $\pi(\cdot|x^e, g)$, then $\forall \pi_1, \pi_2$, we have

$$|J^e(\pi_1) - J^e(\pi_2)| \leq \frac{2\gamma}{1-\gamma} \mathbb{E}_{\rho_{\pi_1}^e(x^e, g)} \left[ D_{TV}(\pi_1(\cdot|x^e, g) \| \pi_2(\cdot|x^e, g)) \right]$$

*Proof.* By the definition of $J^e(\pi)$, we have

$$\begin{aligned}
|J^e(\pi_1) - J^e(\pi_2)| &= |\mathbb{E}_{g \sim \mathcal{G}}[p_{\pi_1}^{\Delta,e}(g|g) - p_{\pi_2}^{\Delta,e}(g|g)]| \\
&\leq \mathbb{E}_{g \sim \mathcal{G}}[|p_{\pi_1}^{\Delta,e}(g|g) - p_{\pi_2}^{\Delta,e}(g|g)|]
\end{aligned}$$

Thus, it suffices to prove $\forall g \in \mathcal{G}$,

$$|p_{\pi_1}^{\Delta,e}(g|g) - p_{\pi_2}^{\Delta,e}(g|g)| \leq \frac{2\gamma}{1-\gamma}\mathbb{E}_{\rho_{\pi_1}^e(x^e|g)}\left[D_{TV}(\pi_1(\cdot|x^e,g) \parallel \pi_2(\cdot|x^e,g))\right]$$

First, we consider $|p_{\pi_1}^e(s_T = s|g) - p_{\pi_2}^e(s_T = s|g)|$ for some fixed step $T$. Denote $\{\pi_1 < t, \pi_2 \geq t\}$ as another policy which imitates policy $\pi_1$ for first $t$ steps and then imitates $\pi_2$ for the rest. By the telescoping operation, we have $\forall s \in \mathcal{S}, g \in \mathcal{G}$

$$|p_{\pi_1}^e(s_T = s|g) - p_{\pi_2}^e(s_T = s|g)| \leq \sum_{t=0}^{T-1}|p_{\pi_1<t,\pi_2\geq t}^e(s_T = s|g) - p_{\pi_1<t+1,\pi_2\geq t+1}^e(s_T = s|g)|$$

$$= \sum_{t=0}^{T-1} P_t^e(s|\pi_1,\pi_2,g,T)$$

where we use $P_t^e(s|\pi_1,\pi_2,g,T)$ to denote each term for brevity.

$$P_t^e(s|\pi_1,\pi_2,g,T) = |\int_{s_t,a} p_{\pi_1}^e(s_t|g)\pi_2(a|x^e(s_t),g)p_{\pi_2}^e(s_T = s|s_t,a,g)ds_tda-$$

$$-\int_{s_t,a} p_{\pi_1}^e(s_t|g)\pi_1(a|x_t^e(s_t),g)p_{\pi_2}^e(s_T = s|s_t,a,g)ds_tda|$$

$$\leq \int_{s_t,a} p_{\pi_1}^e(s_t|g)|\pi_1(a|x^e(s_t),g) - \pi_2(a|x^e(s_t),g)|p_{\pi_2}^e(s_T = s|s_t,a,g)ds_tda$$

$$\leq \int_{s_t} p_{\pi_1}^e(s_t|g)\left(\int_a |\pi_1(a|x^e(s_t),g) - \pi_2(a|x^e(s_t),g)|da\right)ds_t$$

$$\leq 2\int_{s_t} p_{\pi_1}^e(s_t|g)D_{TV}(\pi_1(\cdot|x^e(s_t),g) \parallel \pi_2(\cdot|x^e(s_t),g))ds_t$$

Here, $p_{\pi_2}^e(s_T = s|s_t,a,g)$ is the probability of achieving state $s$ at step $T$ under policy $\pi_2(\cdot|x^e,g)$ when it takes action $a$ at $s_t$. Noticing that, the upper bound for $P_t^e(s|\pi_1,\pi_2,g,T)$ is not dependent on $T$. Thus, we have $\forall s \in \mathcal{S}, g \in \mathcal{G}$,

$$|p_{\pi_1}^{\Delta,e}(s|g) - p_{\pi_2}^{\Delta,e}(s|g)| \leq (1-\gamma)\sum_{T=0}^{\infty}\gamma^T \sum_{t=0}^{T-1} P_t^e(s|\pi_1,\pi_2,g,T)$$

$$= (1-\gamma)\sum_{t=0}^{\infty}\frac{\gamma^{t+1}}{1-\gamma}P_t^e(s|\pi_1,\pi_2,g,t+1)$$

$$\leq 2\sum_{t=0}^{\infty}\gamma^{t+1}\int_{s_t} p_{\pi_1}^e(s_t|g)D_{TV}(\pi_1(\cdot|x^e(s_t),g) \parallel \pi_2(\cdot|x^e(s_t),g))ds_t$$

$$= 2\gamma\int_{s_t}\sum_{t=0}^{\infty}\gamma^t p_{\pi_1}^e(s_t|g)D_{TV}(\pi_1(\cdot|x^e(s_t),g) \parallel \pi_2(\cdot|x^e(s_t),g))ds_t$$

$$= \frac{2\gamma}{1-\gamma}\mathbb{E}_{\rho_{\pi_1}^e(x^e|g)}\left[D_{TV}(\pi_1(\cdot|x^e,g) \parallel \pi_2(\cdot|x^e,g)\right]$$

The Lemma holds by averaging over $g \sim \mathcal{G}$. $\qquad\square$

Lemma 1 bounds the objective function between two policies $\pi_1$ and $\pi_2$ with the Total Variation distance. Recall that we use $\epsilon^{\rho(x,g)}(\pi_1 \parallel \pi_2)$ to denote the average $D_{\mathrm{TV}}$ between the two policies under the joint distribution. We refer to $\pi_G$ as some optimal and invariant policy. Then, we have the following Lemma.

**Lemma 2.** *For any policy class $\Pi$ and two joint distributions $\rho^s(x,g)$ and $\rho^t(x,g)$, suppose*

$$\pi_{s,t}^* = \arg\min_{\pi'\in\Pi} \epsilon^{\rho^s(x,g)}(\pi' \parallel \pi_G) + \epsilon^{\rho^t(x,g)}(\pi' \parallel \pi_G)$$

*then we have for any $\pi \in \Pi$*

$$\epsilon^{\rho^t(x,g)}(\pi \parallel \pi_G) \leq \epsilon^{\rho^s(x,g)}(\pi \parallel \pi_G) + \underbrace{\sup_{\pi,\pi' \in \Pi} \left| \epsilon^{\rho^s(x,g)}(\pi \parallel \pi') - \epsilon^{\rho^t(x,g)}(\pi \parallel \pi') \right|}_{d_{\Pi\Delta\Pi}(\rho^s(x,g),\rho^t(x,g))} + \lambda_{s,t}$$

*where $\lambda_{s,t} = \epsilon^{\rho^s(x,g)}(\pi_{s,t}^* \parallel \pi_G) + \epsilon^{\rho^t(x,g)}(\pi_{s,t}^* \parallel \pi_G)$.*

*Proof.* Noticing that $D_{\text{TV}}$ is a distance metric that satisfies the triangular inequality and is symmetric. Thus, we have $\forall \pi_1, \pi_2, \pi_3$ and any joint distribution $\rho(x,g)$.

$$\begin{aligned}
\epsilon^{\rho(x,g)}(\pi_1 \parallel \pi_2) &= \mathbb{E}_{\rho(x,g)}[D_{\text{TV}}(\pi_1(\cdot|x,g) \parallel \pi_2(\cdot|x,g))] \\
&\leq \mathbb{E}_{\rho(x,g)}[D_{\text{TV}}(\pi_1(\cdot|x,g) \parallel \pi_3(\cdot|x,g)) + D_{\text{TV}}(\pi_3(\cdot|x,g) \parallel \pi_2(\cdot|x,g))] \\
&= \epsilon^{\rho(x,g)}(\pi_1 \parallel \pi_3) + \epsilon^{\rho(x,g)}(\pi_3 \parallel \pi_2)
\end{aligned}$$

Based on this property, we have

$$\begin{aligned}
\epsilon^{\rho^t(x,g)}(\pi \parallel \pi_G) &\leq \epsilon^{\rho^t(x,g)}(\pi_{s,t}^* \parallel \pi_G) + \epsilon^{\rho^t(x,g)}(\pi \parallel \pi_{s,t}^*) \\
&\leq \epsilon^{\rho^t(x,g)}(\pi_{s,t}^* \parallel \pi_G) + \epsilon^{\rho^s(x,g)}(\pi \parallel \pi_{s,t}^*) + \left| \epsilon^{\rho^t(x,g)}(\pi \parallel \pi_{s,t}^*) - \epsilon^{\rho^s(x,g)}(\pi \parallel \pi_{s,t}^*) \right| \\
&\leq \epsilon^{\rho^s(x,g)}(\pi \parallel \pi_G) + \epsilon^{\rho^s(x,g)}(\pi_{s,t}^* \parallel \pi_G) + \epsilon^{\rho^t(x,g)}(\pi_{s,t}^* \parallel \pi_G) \\
&\quad + \left| \epsilon^{\rho^t(x,g)}(\pi \parallel \pi_{s,t}^*) - \epsilon^{\rho^s(x,g)}(\pi \parallel \pi_{s,t}^*) \right| \\
&\leq \epsilon^{\rho^s(x,g)}(\pi \parallel \pi_G) + d_{\Pi\Delta\Pi}(\rho^s(x,g),\rho^t(x,g)) + \lambda_{s,t}
\end{aligned}$$

$\square$

Lemma 2 resembles the seminal bound in domain adaptation theory Ben-David et al. [2010]. Then, we extend the generalization bound to the GBMDP setting. We consider the joint distributions $\rho^E = \{\rho^{e_i}(x,g)\}_{i=1}^N$ and define the characteristic set as follows. [2]

**Definition 2.** *The characteristic set $B(\delta, E|\Pi)$ is a set of joint distributions $\rho(x,g)$ which $\forall e_i \in E$ with $\rho^{e_i}(x,g)$,*

$$d_{\Pi\Delta\Pi}(\rho(x,g), \rho^{e_i}(x,g)) \leq \delta$$

In other words, the characteristic set is a set of joint distributions that is close to the training environments' distributions in terms of the $d_{\Pi\Delta\Pi}$ divergence. Notice that if we define $\delta_E = \max_{e,e' \in E} d_{\Pi\Delta\Pi}(\rho^e(x,g), \rho^{e'}(x,g))$, then we have that the convex hull $\Lambda(\{\rho^{e_i}(x,g)\}_{i=1}^N) \subset B(\delta_E, E|\Pi)$. Namely, the characteristic set contains all the distributions of convex combinations of training environments' distributions Sicilia et al. [2021].

**Proposition 3.** *For any policy class $\Pi$, a set of source joint distributions $\rho^E = \{\rho^{e_i}(x,g)\}_{i=1}^N$ and the target distribution $\rho^t(x,g)$, suppose for any unit sum weights $\{\alpha_i\}_{i=1}^N$, i.e., $0 \leq \alpha_i \leq 1, \sum_i \alpha_i = 1$.*

$$\lambda_\alpha = \sum_{i=1}^N \alpha_i \epsilon^{e_i}(\pi_\alpha^* \parallel \pi_G) + \epsilon^t(\pi_\alpha^* \parallel \pi_G), \quad \pi_\alpha^* = \arg\min_{\pi' \in \Pi} \sum_{i=1}^N \alpha_i \epsilon^{e_i}(\pi' \parallel \pi_G) + \epsilon^t(\pi' \parallel \pi_G)$$

*where $\epsilon^{e_i}$ and $\epsilon^t$ are short of $\epsilon^{\rho^{e_i}}$ and $\epsilon^{\rho^t}$ respectively. Let*

$$\tilde{\rho}(x,g) = \arg\min_{\rho \in B(\delta_E, E|\Pi)} d_{\Pi\Delta\Pi}(\rho(x,g), \rho^t(x,g))$$

*Then, $\forall \pi \in \Pi$*

$$\epsilon^t(\pi \parallel \pi_G) \leq \min_\alpha \sum_{i=1}^N \alpha_i \epsilon^{e_i}(\pi \parallel \pi_G) + \lambda_\alpha + \underbrace{\max_{e,e' \in E} d_{\Pi\Delta\Pi}(\rho^e(x,g), \rho^{e'}(x,g))}_{\delta_E}$$
$$+ d_{\Pi\Delta\Pi}(\tilde{\rho}(x,g), \rho^t(x,g))$$

---

[2] $e_i$ is only used as an index here.

*Proof.* By Lemma 2, we have $\forall e_i \in E$

$$\epsilon^t(\pi \parallel \pi_G) \le \epsilon^{e_i}(\pi \parallel \pi_G) + d_{\Pi\Delta\Pi}(\rho^{e_i}(x,g), \rho^t(x,g)) + \lambda_{e_i,t}$$

Then, we have for any unit sum weights $\alpha$

$$\epsilon^t(\pi \parallel \pi_G) \le \sum_{i=1}^{N} \alpha_i \epsilon^{e_i}(\pi \parallel \pi_G) + \sum_{i=1}^{N} \alpha_i d_{\Pi\Delta\Pi}(\rho^{e_i}(x,g), \rho^t(x,g)) + \sum_{i=1}^{N} \alpha_i \lambda_{e_i,t}$$

Noticing that

$$\sum_{i=1}^{N} \alpha_i \lambda_{e_i,t} = \sum_{i=1}^{N} \alpha_i \left( \min_{\pi' \in \Pi} \epsilon^{e_i}(\pi' \parallel \pi_G) + \epsilon^t(\pi' \parallel \pi_G) \right)$$

$$\le \min_{\pi' \in \Pi} \sum_{i=1}^{N} \alpha_i \epsilon^{e_i}(\pi' \parallel \pi_G) + \epsilon^t(\pi' \parallel \pi_G)$$

$$= \lambda_\alpha$$

Thus, we have

$$\epsilon^t(\pi \parallel \pi_G) \le \sum_{i=1}^{N} \alpha_i \epsilon^{e_i}(\pi \parallel \pi_G) + \sum_{i=1}^{N} \alpha_i d_{\Pi\Delta\Pi}(\rho^{e_i}(x,g), \rho^t(x,g)) + \lambda_\alpha$$

Since $d_{\Pi\Delta\Pi}$ divergence also follows the triangular inequality Ben-David et al. [2010], we have

$$\epsilon^t(\pi \parallel \pi_G) \le \sum_{i=1}^{N} \alpha_i \epsilon^{e_i}(\pi \parallel \pi_G) + \sum_{i=1}^{N} \alpha_i d_{\Pi\Delta\Pi}(\rho^{e_i}(x,g), \tilde{\rho}(x,g)) + d_{\Pi\Delta\Pi}(\tilde{\rho}(x,g), \rho^t(x,g)) + \lambda_\alpha$$

$$\le \sum_{i=1}^{N} \alpha_i \epsilon^{e_i}(\pi \parallel \pi_G) + \lambda_\alpha + \delta_E + d_{\Pi\Delta\Pi}(\tilde{\rho}(x,g), \rho^t(x,g))$$

The proposition holds by taking the minimum over $\alpha$. □

Finally, we are able to provide the formal statements and proofs for Proposition 1 as follows.

**Proposition 1** (Formal). *For any $\pi \in \Pi$, we consider the occupancy measure $\rho^{\mathcal{E}_{train}} = \{\rho_\pi^{e_i}(x^{e_i}, g)\}_{i=1}^{N}$ for training environments and $\rho_{\pi_G}^t(x^t, g)$ for the target environment. For simplicity, we use $\epsilon^{e_i}$ and $\epsilon^t$ as the abbreviations. Considering*

$$\lambda = \frac{1}{N} \sum_{i=1}^{N} \epsilon^{e_i}(\pi^* \parallel \pi_G) + \epsilon^t(\pi^* \parallel \pi_G), \quad \pi^* = \arg\min_{\pi' \in \Pi} \sum_{i=1}^{N} \epsilon^{e_i}(\pi' \parallel \pi_G)$$

*and $\delta = \max_{e,e' \in \mathcal{E}_{train}} d_{\Pi\Delta\Pi}(\rho_\pi^e(x^e, g), \rho_\pi^{e'}(x^{e'}, g))$, the characteristic set $B(\delta, \mathcal{E}_{train}|\Pi)$. Define*

$$\tilde{\rho}(x,g) = \arg\min_{\rho \in B(\delta, \mathcal{E}_{train}|\Pi)} d_{\Pi\Delta\Pi}(\rho(x,g), \rho_{\pi_G}^t(x^t, g))$$

*Then, we have*

$$J^t(\pi_G) - J^t(\pi) \le \frac{2\gamma}{1-\gamma} \left[ \frac{1}{N} \sum_{i=1}^{N} \epsilon^{e_i}(\pi \parallel \pi_G) + \lambda + \delta + d_{\Pi\Delta\Pi}(\tilde{\rho}(x,g), \rho_{\pi_G}^t(x^t, g)) \right]$$

*Proof.* By Lemma 1 and Proposition 3, we have

$$J^t(\pi_G) - J^t(\pi) \le \frac{2\gamma}{1-\gamma} \epsilon^t(\pi \parallel \pi_G)$$

$$\le \frac{2\gamma}{1-\gamma} \left[ \min_\alpha \sum_{i=1}^{N} \alpha_i \epsilon^{e_i}(\pi \parallel \pi_G) + \lambda_\alpha + \delta + d_{\Pi\Delta\Pi}(\tilde{\rho}(x,g), \rho_{\pi_G}^t(x,g)) \right]$$

$$\leq \frac{2\gamma}{1-\gamma}\left[\frac{1}{N}\sum_{i=1}^{N}\epsilon^{e_i}(\pi \parallel \pi_G) + \lambda_{\alpha=\frac{1}{N}} + \delta + d_{\Pi\Delta\Pi}(\tilde{\rho}(x,g), \rho^t_{\pi_G}(x,g))\right]$$

Noticing that $\lambda$ and $\lambda_{\alpha=\frac{1}{N}}$ have different definitions. But, we have

$$\lambda_{\alpha=\frac{1}{N}} = \min_{\pi'\in\Pi}\frac{1}{N}\sum_{i=1}^{N}\epsilon^{e_i}(\pi' \parallel \pi_G) + \epsilon^t(\pi' \parallel \pi_G)$$

$$\leq \frac{1}{N}\sum_{i=1}^{N}\epsilon^{e_i}(\pi^* \parallel \pi_G) + \epsilon^t(\pi^* \parallel \pi_G)$$

$$= \lambda$$

where $\pi^* = \arg\min_{\pi'\in\Pi}\sum_{i=1}^{N}\epsilon^{e_i}(\pi' \parallel \pi_G)$. Thus, we have

$$J^t(\pi_G) - J^t(\pi) \leq \frac{2\gamma}{1-\gamma}\left[\frac{1}{N}\sum_{i=1}^{N}\epsilon^{e_i}(\pi \parallel \pi_G) + \lambda + \delta + d_{\Pi\Delta\Pi}(\tilde{\rho}(x,g), \rho^t_{\pi_G}(x^t,g))\right]$$

This completes the proof. □

**Remark 1.** *The informal version Proposition 1 omits unessential constant part and the definition of the characteristic set.*

## C.3  Proof of Proposition 2

Here, we provide the formal proof and statement of Proposition 2. To begin with, we prove the following Lemma.

**Lemma 3.** *For two goal-conditioned policies $\pi, \pi'$ of the Goal-conditioned MDP $\langle \mathcal{S}, \mathcal{G}, \mathcal{A}, p, \gamma \rangle$, suppose that $\max_{s,g} D_{TV}(\pi(\cdot|s,g) \parallel \pi'(\cdot|s,g)) \leq \epsilon$, then we have $D_{TV}(\rho^\pi(s,g) \parallel \rho^{\pi'}(s,g)) \leq \frac{\gamma\epsilon}{1-\gamma}$.*

*Proof.* The proof follows the perturbation theory in Appendix B of Schulman et al. [2015]. By the definition of total variation distance, it suffices to prove that $\forall g, D_{TV}(\rho^\pi(s|g) \parallel \rho^{\pi'}(s|g)) \leq \frac{\gamma\epsilon}{1-\gamma}$, where $\rho^\pi(s|g)$ denotes the discounted occupancy measure of $s$ under policy $\pi(\cdot|s,g)$. Consequently, in the following notations, we may omit specifying $g$ if unambiguous.

First, we refer $P_\pi$ as the state transition matrix under policy $\pi(\cdot|s,g)$, i.e., $(P_\pi)_{xy} = \int_a p(s' = x|s = y, a)\pi(a|s = y, g)da$ and subsequently, $G_\pi = I + \gamma P_\pi + \gamma^2 P_\pi^2 + \cdots = (I - \gamma P_\pi)^{-1}$. Then, the transition discrepancy matrix is defined as $\Delta = P_{\pi'} - P_\pi$. Observing that

$$G_\pi^{-1} - G_{\pi'}^{-1} = \gamma(P_{\pi'} - P_\pi)$$
$$= \gamma\Delta$$
$$\Rightarrow G_{\pi'} - G_\pi = \gamma G_{\pi'}\Delta G_\pi$$

Thus, for any initial state distribution $\rho_0$, we have

$$D_{TV}(\rho^\pi(s|g) \parallel \rho^{\pi'}(s|g)) = \frac{1}{2}\sum_s |\rho^\pi(s|g) - \rho^{\pi'}(s|g)|$$

$$= \frac{1-\gamma}{2} \parallel (G_\pi - G_{\pi'})\rho_0 \parallel_1$$

$$= \frac{(1-\gamma)\gamma}{2} \parallel G_{\pi'}\Delta G_\pi \rho_0 \parallel_1$$

$$\leq \frac{(1-\gamma)\gamma}{2} \parallel G_{\pi'} \parallel_1 \parallel \Delta \parallel_1 \parallel G_\pi \parallel_1 \parallel \rho_0 \parallel_1$$

Noticing that $P_\pi, P_{\pi'}$ are matrices whose columns have sum 1. Thus, $\parallel G_\pi \parallel_1 = \parallel G_{\pi'} \parallel_1 = \frac{1}{1-\gamma}$. Furthermore,

$$\parallel \Delta \parallel_1 = \max_y \int_x \left|\int_a p(s' = x|s = y, a)(\pi'(a|s = y, g) - \pi(a|s = y, g))da\right| dx$$

$$= \max_y \int_a \int_x p(s' = x | s = y, a) \, |\pi'(a | s = y, g) - \pi(a | s = y, g)| \, dx da$$

$$= \max_y 2 D_{\text{TV}}(\pi(\cdot | s = y, g) \parallel \pi'(\cdot | s = y, g))$$

$$\leq 2\epsilon$$

In all, we have

$$D_{\text{TV}}(\rho^\pi(s|g) \parallel \rho^{\pi'}(s|g)) \leq \frac{\gamma \epsilon}{1 - \gamma}$$

$\square$

To state Proposition 2 formally, we define the $L$-lipschitz policy class $\Pi_{\Phi,L}^E$, whose $\Phi : \mathcal{X}^E \to \mathcal{Z}$ maps the input $x^e$s to latent vector $z$s. [3]

$$\Pi_{\Phi,L}^E = \{w(\Phi(x^e), g), \forall w | \forall g \in \mathcal{G}, z, z' \in \Phi(\mathcal{X}^E), \ D_{\text{TV}}(w(z,g) \parallel w(z',g)) \leq L \parallel z - z' \parallel_2\}$$

Namely, the nonlinear function $w$ is $L$-smooth over the latent space $\Phi(\mathcal{X}^E)$ for each $g$. Furthermore, we extend the definition of perfect alignment encoder to the $(\eta, \psi)$-*perfect alignment* as follows.

**Definition 3** ($(\eta, \psi)$-**Perfect Alignment**). *An encoder $\Phi$ is a $(\eta, \psi)$-perfect alignment encoder over the environments $E$, if it satisfies the following two properties:*

1. $\forall s \in \mathcal{S}, \forall e, e' \in E, \parallel \Phi(x^e(s)) - \Phi(x^{e'}(s)) \parallel_2 \leq \eta.$

2. $\forall s, s' \in \mathcal{S}, \forall e, e' \in E, \parallel \Phi(x^e(s)) - \Phi(x^{e'}(s')) \parallel_2 \geq \psi \parallel s - s' \parallel_2$

Essentially, $\eta$ quantifies the *if* condition of perfect alignment (Definition 1), i.e., how aligned the representation $\Phi(x^e(s)), e \in E$s are. Moreover, $\psi$ quantifies the *only if* condition, i.e., how the representations of different states $s$ are separated. Based on the definition of $(\eta, \psi)$-perfect alignment, we prove the formal statement of Proposition 2 as follows.

**Proposition 2** (Formal). *$\forall \pi \in \Pi_{\Phi,L}^{\mathcal{E}_{train}}$ and occupancy measure $\rho^{\mathcal{E}_{train}} = \{\rho_\pi^{e_i}(x^{e_i}, g)\}_{i=1}^N$ for training environments and $\rho_{\pi_G}^t(x^t, g)$ for the target environment. For simplicity, we use $\epsilon^{e_i}$ and $\epsilon^t$ as the abbreviations. Considering $\pi^* = \arg\min_{\pi' \in \Pi_{\Phi,L}^{\mathcal{E}_{train}}} \sum_{i=1}^N \epsilon^{e_i}(\pi' \parallel \pi_G)$ and*

$$\tilde{\rho}(x, g) = \arg\min_{\rho \in B(\delta, \mathcal{E}_{train} | \Pi_{\Phi,L}^{\mathcal{E}_{train}})} d_{\Pi_{\Phi,L}^{\mathcal{E}_{train}} \Delta \Pi_{\Phi,L}^{\mathcal{E}_{train}}}(\rho(x, g), \rho_{\pi_G}^t(x^t, g))$$

*Then, if the encoder $\Phi$ is a $(\eta, \psi)$-perfect alignment over $\mathcal{E}_{train}$ and $\pi_G$ is a $u$-smooth invariant policy with $u = L\psi$, i.e., $\forall s, s', \forall g, D_{\text{TV}}(\pi_G(\cdot | x^e(s), g) \parallel \pi_G(\cdot | x^e(s'), g)) \leq u \parallel s - s' \parallel_2$, we have*

$$J^t(\pi_G) - J^t(\pi) \leq \frac{2\gamma}{1-\gamma} \left[ \underbrace{\frac{1}{N} \sum_{i=1}^N \epsilon^{e_i}(\pi \parallel \pi_G)}_{(E)} + (3 + \frac{\gamma}{1-\gamma})\eta L \right]$$

$$+ \frac{2\gamma}{1-\gamma} \left[ \underbrace{\epsilon^t(\pi^* \parallel \pi_G) + d_{\Pi_{\Phi,L}^{\mathcal{E}_{train}} \Delta \Pi_{\Phi,L}^{\mathcal{E}_{train}}}(\tilde{\rho}(x, g), \rho_{\pi_G}^t(x^t, g))}_{(t)} \right].$$

*Proof.* It suffices to prove the following two statements with Proposition 1.

(1) $\max_{e_i, e_i' \in \mathcal{E}_{train}} d_{\Pi_{\Phi,L}^{\mathcal{E}_{train}} \Delta \Pi_{\Phi,L}^{\mathcal{E}_{train}}}(\rho_\pi^{e_i}(x^{e_i}, g), \rho_\pi^{e_i'}(x^{e_i'}, g)) \leq (2 + \frac{\gamma}{1-\gamma})\eta L.$

(2) $\frac{1}{N} \sum_{i=1}^N \epsilon^{e_i}(\pi^* \parallel \pi_G) \leq \eta L.$

---

[3] $\mathcal{X}^E = \cup_{e \in E} \mathcal{X}^e$

Proof of (1): By the definition of $\Pi_{\Phi,L}^{\mathcal{E}_{\text{train}}}$, we have $\forall s, g$ and $e, e' \in \mathcal{E}_{\text{train}}$, $D_{\text{TV}}(\pi(\cdot|x^e(s), g) \parallel \pi(\cdot|x^{e'}(s), g)) \leq \eta L$. Without loss of generality, we have $\forall e, e' \in \mathcal{E}_{\text{train}}$,

$$
\begin{aligned}
d_{\Pi_{\Phi,L}^{\mathcal{E}_{\text{train}}} \Delta \Pi_{\Phi,L}^{\mathcal{E}_{\text{train}}}}(\rho_\pi^e(x^e, g), \rho_\pi^{e'}(x^{e'}, g)) &= \sup_{\pi, \pi' \in \Pi_{\Phi,L}^{\mathcal{E}_{\text{train}}}} |\mathbb{E}_{\rho_\pi^e(s,g)}[D_{\text{TV}}(\pi(\cdot|x^e(s), g) \parallel \pi'(\cdot|x^e(s), g))] \\
&\quad - \mathbb{E}_{\rho_\pi^{e'}(s,g)}[D_{\text{TV}}(\pi(\cdot|x^{e'}(s), g) \parallel \pi'(\cdot|x^{e'}(s), g))]| \\
&\leq \sup_{\pi, \pi' \in \Pi_{\Phi,L}^{\mathcal{E}_{\text{train}}}} \mathbb{E}_{\rho_\pi^e(s,g)}[D_{\text{TV}}(\pi(\cdot|x^e(s), g) \parallel \pi(\cdot|x^{e'}(s), g))] \\
&\quad + \sup_{\pi, \pi' \in \Pi_{\Phi,L}^{\mathcal{E}_{\text{train}}}} \mathbb{E}_{\rho_\pi^e(s,g)}[D_{\text{TV}}(\pi'(\cdot|x^e(s), g) \parallel \pi'(\cdot|x^{e'}(s), g))] \\
&\quad + \sup_{\pi, \pi' \in \Pi_{\Phi,L}^{\mathcal{E}_{\text{train}}}} |\mathbb{E}_{\rho_\pi^e(s,g)}[D_{\text{TV}}(\pi(\cdot|x^{e'}(s), g) \parallel \pi'(\cdot|x^{e'}(s), g))] \\
&\quad - \mathbb{E}_{\rho_\pi^{e'}(s,g)}[D_{\text{TV}}(\pi(\cdot|x^{e'}(s), g) \parallel \pi'(\cdot|x^{e'}(s), g))]| \\
&\leq 2\eta L + \sup_{A \in \sigma(\mathcal{S}, \mathcal{G})} |\rho_\pi^e(A) - \rho_\pi^{e'}(A)| \\
&= 2\eta L + D_{\text{TV}}(\rho_\pi^e(s, g) \parallel \rho_\pi^{e'}(s, g))
\end{aligned}
$$

Then, by Lemma 3, we have $D_{\text{TV}}(\rho_\pi^e(s, g) \parallel \rho_\pi^{e'}(s, g)) \leq \frac{\gamma \eta L}{1-\gamma}$. Consequently, we have $\forall e, e' \in \mathcal{E}_{\text{train}}$,

$$
d_{\Pi_{\Phi,L}^{\mathcal{E}_{\text{train}}} \Delta \Pi_{\Phi,L}^{\mathcal{E}_{\text{train}}}}(\rho_\pi^e(x^e, g), \rho_\pi^{e'}(x^{e'}, g)) \leq (2 + \frac{\gamma}{1-\gamma})\eta L
$$

Proof of (2): First, for each $z \in \Phi(\mathcal{X}^{\mathcal{E}_{\text{train}}})$, we assign one $s(z)$ such that $\exists e(z) \in \mathcal{E}_{\text{train}}$, s.t. $\Phi(x^{e(z)}(s(z))) = z$. Then, we choose the $\tilde{w}$ that $\tilde{w}(z, g) = \pi_G(\cdot|s(z), g), \forall z \in \Phi(\mathcal{X}^{\mathcal{E}_{\text{train}}})$. Clearly, $\tilde{w}$ is a mapping of $\Phi(\mathcal{X}^{\mathcal{E}_{\text{train}}}) \to \Pi_{\Phi,\infty}^{\mathcal{E}_{\text{train}}}$. Besides, $\forall z_1, z_2 \in \Phi(\mathcal{X}^{\mathcal{E}_{\text{train}}}), g \in \mathcal{G}$, we have

$$
\begin{aligned}
\parallel \tilde{w}(z_1, g) - \tilde{w}(z_2, g) \parallel_2 &\leq u \parallel s(z_1) - s(z_2) \parallel_2 \\
&\leq \frac{u}{\psi}\psi \parallel s(z_1) - s(z_2) \parallel_2 \\
&\leq \frac{u}{\psi} \parallel \Phi(x^{e(z_1)}(s(z_1))) - \Phi(x^{e(z_2)}(s(z_2))) \parallel_2 \\
&\leq L \parallel z_1 - z_2 \parallel_2
\end{aligned}
$$

Thus, the policy $\tilde{\pi} = \tilde{w}(\Phi(x^e), g) \in \Pi_{\Phi,L}^{\mathcal{E}_{\text{train}}}$. Furthermore, by the definition of $(\eta, \psi)$-perfect alignment, we have

$$
\begin{aligned}
\frac{1}{N}\sum_{i=1}^N \epsilon^{e_i}(\tilde{\pi} \parallel \pi_G) &= \frac{1}{N}\sum_{i=1}^N \mathbb{E}_{\rho_\pi^{e_i}(s,g)}[D_{\text{TV}}(\tilde{\pi}(\cdot|x^{e_i}(s), g) \parallel \pi_G(\cdot|s, g)))] \\
&\leq \frac{1}{N}\sum_{i=1}^N \mathbb{E}_{\rho_\pi^{e_i}(s,g)}[D_{\text{TV}}(\tilde{\pi}(\cdot|x^{e_i}(s), g) \parallel \tilde{w}(\Phi(x^{e(z)}(s)), g)))] \\
&\leq \eta L
\end{aligned}
$$

This completes the proof. $\qquad\square$

**Remark 2.** *If we choose* $\psi = \frac{1}{\sqrt{L}}, \eta = \frac{1}{L^2}$ *and* $u = \sqrt{L}$, *the informal version of Proposition 2 describes the case when* $L \to \infty$ *and omits unessential constants.*

### C.4 Discussions on Eq. (3)

Here, we discuss the remaining terms $(E), (t)$ in the R.H.S of Eq. (3), i.e., upper bound of $J^t(\pi_G) - J^t(\pi)$. Together with the empirical analysis, we show how these terms are reduced by our perfect alignment criterion.

**Discussion on** $(E)$**.** Theoretically speaking, for a $(\eta, \psi)$-perfect alignment encoder, we have $\min(E) \leq \eta L$, as proved in Appendix C.3. Thus, when $\Phi$ is an ideal perfect alignment over $\mathcal{E}_{\text{train}}$, i.e., $\eta = 0, \psi > 0$ and $L \to \infty$, the optimal invariant policy $\pi_G \in \Pi_{\Phi,\infty}^{\mathcal{E}_{\text{train}}}$. In Section 4.2, empirical results demonstrate that $\eta$ is minimized to almost 0 and the reconstruction (Figure 9) demonstrates that the *only if* condition is also satisfied.

Moreover, in GBMDPs with finite states, the perfect alignment encoder $\Phi$ over $\mathcal{E}_{\text{train}}$ maps all training environments to the same goal-conditioned MDP (Figure 6(b)) with state space $\{z(s), s \sim \mathcal{S}\}$. Noticing that the mapping is bijective and share the same actions and rewards with the original problem. Thus, a RL algorithm (e.g. Q-learning Watkins and Dayan [1992]) on the equivalent MDP will converge to an optimal policy $\pi_G$ in the original MDP, i.e., invariant and maximize $J^e, e \in \mathcal{E}_{\text{train}}$.

Empirically, in Table 1, we find that ours PA-SF achieves the near-optimal performance on all $\mathcal{E}_{\text{train}}$ s, i.e., the same performance as a policy trained on a single environment. Therefore, we conclude that $(E)$ term is reduced to almost zero.

**Discussion on** $(t)$**.** In general, it is hard to conduct task-agnostic analysis on $(t)$ term, as discussed in Sicilia et al. [2021]. Moreover, owing to the intractable sup operators, it is almost impossible to measure the $(t)$ term directly in the experiments. Here, we analyze the generalization error term $(t)$ with an upper bound and we find evidence that this upper bound is reduced significantly under our perfect alignment criteria.

Here, we analyze the generalization performance when the policy converges to the optimal policy over $\mathcal{E}_{\text{train}}$, which is the case empirically. Furthermore, we assume that $\forall s \in \mathcal{S}, e \in \mathcal{E}_{\text{train}}, p_\pi^e(s) \geq \epsilon$. Since it has been proven that the relabeled goal distribution of Skew-Fit will converge to $U(\mathcal{S})$ (uniform over the bounded state space) Pong et al. [2020], it is reasonable to assume that each state has non-zero occupancy measure under the well-trained policy $\pi(\cdot|s, g)$. We derive the following proposition with the same notation as in Proposition 2 except that $\Pi_{\Phi,L}^{\mathcal{E}_{\text{train}}}$ is replaced by $\Pi_{\Phi,L}^{\mathcal{E}}$.

**Proposition 4.** *Suppose that $\Phi$ is a $(\eta_t, \psi_t)$-perfect alignment encoder over $\mathcal{E}$ and $\pi_G, \pi \in \Pi_{\Phi,L}^{\mathcal{E}}$. Besides, $\forall s \in \mathcal{S}, e \in \mathcal{E}_{\text{train}}, p_\pi^e(s) \geq \epsilon$. Then, $\forall t \in \mathcal{E}/\mathcal{E}_{\text{train}}$, we have*

$$(t) \leq \frac{2\gamma}{1-\gamma}\frac{N(E)}{\epsilon} + 4\eta_t L + (E)$$

*Consequently, we have*

$$J^t(\pi_G) - J^t(\pi) \leq \frac{4\gamma}{1-\gamma}(1 + \frac{\gamma N}{(1-\gamma)\epsilon})(E) + \frac{(14 - 12\gamma)\gamma}{(1-\gamma)^2}\eta_t L$$

*Proof.* It is the straight forward to check that the statement and the proofs in Proposition 2 also hold under the policy class $\Pi_{\Phi,L}^{\mathcal{E}} \subset \Pi_{\Phi,L}^{\mathcal{E}_{\text{train}}}$ when $\Phi$ is $(\eta_t, \psi_t)$-perfect alignment over $\mathcal{E}$. Namely, $\forall \pi, \pi_G \in \Pi_{\Phi,L}^{\mathcal{E}}$, training environment set $\mathcal{E}_{\text{train}}$ and the target environment $t \in \mathcal{E}/\mathcal{E}_{\text{train}}$, we have

$$J^t(\pi_G) - J^t(\pi) \leq \frac{2\gamma}{1-\gamma}\left[\underbrace{\frac{1}{N}\sum_{i=1}^{N}\epsilon^{e_i}(\pi \parallel \pi_G)}_{(E)} + (3 + \frac{\gamma}{1-\gamma})\eta_t L\right]$$

$$+ \frac{2\gamma}{1-\gamma}\left[\underbrace{\epsilon^t(\pi^* \parallel \pi_G) + d_{\Pi_{\Phi,L}^{\mathcal{E}}\Delta\Pi_{\Phi,L}^{\mathcal{E}}}(\tilde{\rho}(x, g), \rho_{\pi_G}^t(x^t, g))}_{(t)}\right].$$

By definition, $\forall e \in \mathcal{E}_{\text{train}}$, we have $\rho_\pi^e \in B(\delta, \mathcal{E}_{\text{train}}|\Pi_{\Phi,L}^{\mathcal{E}})$. As a consequence, we have

$$(t) = \frac{1}{N}\sum_{i=1}^{N}(\epsilon^t(\pi^* \parallel \pi_G) - \epsilon^{e_i}(\pi^* \parallel \pi_G)) + \frac{1}{N}\sum_{i=1}^{N}\epsilon^{e_i}(\pi^* \parallel \pi_G)$$

$$+ d_{\Pi_{\Phi,L}^{\mathcal{E}}\Delta\Pi_{\Phi,L}^{\mathcal{E}}}(\tilde{\rho}(x, g), \rho_{\pi_G}^t(x^t, g))$$

$$\leq \max_{e\in\mathcal{E}_{\text{train}}} \epsilon^t(\pi^* \parallel \pi_G) - \epsilon^e(\pi^* \parallel \pi_G) + d_{\Pi^{\mathcal{E}}_{\Phi,L}\Delta\Pi^{\mathcal{E}}_{\Phi,L}}(\rho^e_\pi(x,g), \rho^t_{\pi_G}(x^t,g)) + \lambda$$

$$\leq 2 \max_{e\in\mathcal{E}_{\text{train}}} d_{\Pi^{\mathcal{E}}_{\Phi,L}\Delta\Pi^{\mathcal{E}}_{\Phi,L}}(\rho^e_\pi(x^e,g), \rho^t_{\pi_G}(x^t,g)) + \lambda$$

$$\leq \lambda + 2 \max_{e\in\mathcal{E}_{\text{train}}} d_{\Pi^{\mathcal{E}}_{\Phi,L}\Delta\Pi^{\mathcal{E}}_{\Phi,L}}(\rho^e_\pi(x^e,g), \rho^e_{\pi_G}(x^e,g))$$

$$+ 2 \max_{e\in\mathcal{E}_{\text{train}}} d_{\Pi^{\mathcal{E}}_{\Phi,L}\Delta\Pi^{\mathcal{E}}_{\Phi,L}}(\rho^e_{\pi_G}(x^e,g), \rho^t_{\pi_G}(x^t,g))$$

Without loss of generality, we have

$$d_{\Pi^{\mathcal{E}}_{\Phi,L}\Delta\Pi^{\mathcal{E}}_{\Phi,L}}(\rho^e_{\pi_G}(x^e,g), \rho^t_{\pi_G}(x^t,g)) = \sup_{\pi,\pi'\in\Pi^{\mathcal{E}}_{\Phi,L}} |\mathbb{E}_{\rho^e_{\pi_G}(s,g)}[D_{\text{TV}}(\pi(\cdot|x^e(s),g) \parallel \pi'(\cdot|x^e(s),g))]$$

$$- \mathbb{E}_{\rho^t_{\pi_G}(s,g)}[D_{\text{TV}}(\pi(\cdot|x^t(s),g) \parallel \pi'(\cdot|x^t(s),g))]|$$

$$\leq \mathbb{E}_{\rho_{\pi_G}(s,g)}[\sup_{\pi,\pi'\in\Pi^{\mathcal{E}}_{\Phi,L}} |D_{\text{TV}}(\pi(\cdot|x^e(s),g) \parallel \pi'(\cdot|x^e(s),g))$$

$$- D_{\text{TV}}(\pi(\cdot|x^t(s),g) \parallel \pi'(\cdot|x^t(s),g))|]$$

$$\leq \mathbb{E}_{\rho_{\pi_G}(s,g)}[\sup_{\pi,\pi'\in\Pi^{\mathcal{E}}_{\Phi,L}} |D_{\text{TV}}(\pi(\cdot|x^e(s),g) \parallel \pi(\cdot|x^t(s),g))$$

$$+ D_{\text{TV}}(\pi(\cdot|x^t(s),g) \parallel \pi'(\cdot|x^t(s),g))$$

$$+ D_{\text{TV}}(\pi'(\cdot|x^t(s),g) \parallel \pi'(\cdot|x^e(s),g))$$

$$- D_{\text{TV}}(\pi(\cdot|x^t(s),g) \parallel \pi'(\cdot|x^t(s),g))|]$$

$$\leq \mathbb{E}_{\rho_{\pi_G}(s,g)}[\sup_{\pi,\pi'\in\Pi^{\mathcal{E}}_{\Phi,L}} |D_{\text{TV}}(\pi(\cdot|x^e(s),g) \parallel \pi(\cdot|x^t(s),g))$$

$$+ D_{\text{TV}}(\pi'(\cdot|x^t(s),g) \parallel \pi'(\cdot|x^e(s),g))|]$$

Clearly, we have $\forall e,e' \in \mathcal{E}, D_{\text{TV}}(\pi(\cdot|x^e(s),g) \parallel \pi(\cdot|x^{e'}(s),g)) \leq \eta_t L$ and $\forall e,e' \in \mathcal{E}_{\text{train}}, D_{\text{TV}}(\pi(\cdot|x^e(s),g) \parallel \pi_G(\cdot|x^e(s),g)) \leq \frac{N(E)}{\epsilon}$. Thus, by Lemma 3, we have $\forall e \in \mathcal{E}_{\text{train}}, D_{\text{TV}}(\rho^e_\pi(s,g) \parallel \rho^e_{\pi_G}(s,g)) \leq \frac{\gamma}{1-\gamma}\frac{N(E)}{\epsilon}$. By the fact that $\lambda \leq (E)$, we have

$$(t) \leq \frac{2\gamma}{1-\gamma}\frac{N(E)}{\epsilon} + 4\eta_t L + (E)$$

Consequently, we have

$$J^t(\pi_G) - J^t(\pi) \leq \frac{4\gamma}{1-\gamma}(1 + \frac{\gamma N}{(1-\gamma)\epsilon})(E) + \frac{(14-12\gamma)\gamma}{(1-\gamma)^2}\eta_t L$$

$\square$

Intuitively speaking, when the RL policy converges to nearly optimal on $\mathcal{E}_{\text{train}}$, the generalization regret over target environment can be bounded by the sum of two components: training environment performance regret and the level of invariant over the target environment. The former is relative small as the policy is near optimal. The latter can be reduced by learning perfect alignment encoder on the training environments. As shown in Figure 4, the LER of different ablations over test environments suggests that $\eta_t$ is reduced empirically in PA-SF. Moreover, both losses $\mathcal{L}_{\text{MMD}}$ and $\mathcal{L}_{\text{DIFF}}$ contribute to the reduction. Furthermore, we notice that PA-SF < PA-SF (w/o D) < PA-SF (w/o MD) < Skew-Fit in both LER and generalization performance, which coincides with our theoretical analysis.

## C.5 Discussions on $L^{\text{MMD}}$

Noticing that the $(\eta,\psi)$-perfect alignment requires the $l_2$ distance of $z^e(s) - z^{e'}(s)$ is bounded for any state $s$. In practice, we adopt the convention Schulman et al. [2015] to minimize the average $l_2$ distance over the state space as a surrogate objective, i.e., $\mathbb{E}_{\rho(s)}[\parallel \Phi(x^e(s)) - \Phi(x^{e'}(s)) \parallel^2_2]$. It prevents the unstable and intractable training of robust optimization in our setting and we find it works well empirically. Here, we introduce a theoretical property of $L^{\text{MMD}}$, which justifies its validity to ensure the *if* condition of perfect alignment. To begin with, we prove the following Lemma.

**Lemma 4.** *For some distribution $P(x)$ and two functions $f, g : \mathcal{X} \to \mathbb{R}^d$, we have*

$$\mathbb{E}_{x,x'\sim P(x)}[\| f(x) - g(x') \|_2^2] \geq \frac{1}{2}\mathbb{E}_{x\sim P(x)}[\| f(x) - g(x) \|_2^2]$$

*Proof.* Without loss of generality, we assume $\mathbb{E}_{x\sim P(x)}[g(x)] = 0$. Then, we have

$$
\begin{aligned}
\mathbb{E}_{x,x'\sim P(x)}[\| f(x) - g(x') \|_2^2] &= \mathbb{E}_{x\sim P(x)}[\| f(x) \|_2^2] + \mathbb{E}_{x'\sim P(x')}[\| g(x') \|_2^2] \\
&\quad - 2\mathbb{E}_{x,x'\sim P(x)}[\langle f(x), g(x')\rangle] \\
&= \mathbb{E}_{x\sim P(x)}[\| f(x) \|_2^2] + \mathbb{E}_{x'\sim P(x')}[\| g(x') \|_2^2] \\
&\quad - 2\langle \mathbb{E}_{x\sim P(x)}[f(x)], \mathbb{E}_{x'\sim P(x')}[g(x')]\rangle \\
&= \mathbb{E}_{x\sim P(x)}[\| f(x) \|_2^2] + \mathbb{E}_{x'\sim P(x')}[\| g(x') \|_2^2] \\
&\geq \frac{1}{2}\mathbb{E}_{x\sim P(x)}[\| f(x) - g(x) \|_2^2]
\end{aligned}
$$

$\square$

**Proposition 5.** *Under mild assumptions, if $\mathbb{E}_{e,e'\sim\mathcal{E}_{train},\mathcal{B}_{align}\sim\mathcal{R}_{align}}[\| \Phi(x^e(s)) - \Phi(x^{e'}(s)) \|_2^2] \geq \delta$, then $L^{MMD}(\Phi) \geq O(\delta)$.*

*Proof.* We assume that $\forall z, z', \| \psi(z) - \psi(z') \|_2 \geq u \| z - z' \|_2$ for some $u > 0$. For brevity, we denote $z^e(s) = \Phi(x^e(s))$ and $s_b^e = s_t^e(a_{0:t}^b)$ (the $b^{th}$ sample: state $s$ sampled in environment $e$ after executing some action $a_{0:t}^b$).

$$
\begin{aligned}
L^{MMD}(\Phi) &= \mathbb{E}_{e,e'\sim\mathcal{E}_{train},\mathcal{B}_{align}\sim\mathcal{R}_{align}}[\| \frac{1}{B}\sum_{b=1}^{B}\psi(z^e(s_b^e)) - \frac{1}{B}\sum_{b=1}^{B}\psi(z^{e'}(s_b^{e'})) \|_2^2] \\
&\geq \frac{1}{B^2}\mathbb{E}_{e,e'\sim\mathcal{E}_{train},\{s_b^e,s_b^{e'}\}_{b=1}^B\sim\mathcal{R}_{align}}[\sum_{b=1}^{B}\| \psi(z^e(s_b^e)) - \psi(z^{e'}(s_b^{e'})) \|_2^2] + \frac{1}{B^2} \\
&\quad \mathbb{E}_{e,e'\sim\mathcal{E}_{train},\{s_b^e,s_b^{e'}\}_{b=1}^B\sim\mathcal{R}_{align}}[\sum_{i\neq j}\langle\psi(z^e(s_i^e)) - \psi(z^{e'}(s_i^{e'})), \psi(z^e(s_j^e)) - \psi(z^{e'}(s_j^{e'}))\rangle]
\end{aligned}
$$

Since $s_b^e$s are sampled from $\mathcal{R}_{align}$ independently, we have

$$
\begin{aligned}
L^{MMD}(\Phi) &\geq \frac{1}{B}\mathbb{E}_{e,e'\sim\mathcal{E}_{train},s_b^e,s_b^{e'}\sim\mathcal{R}_{align}}[\| \psi(z^e(s_b^e)) - \psi(z^{e'}(s_b^{e'})) \|_2^2] \\
&\quad + \frac{B-1}{B}\| \mathbb{E}_{e,e'\sim\mathcal{E}_{train},s_b^e,s_b^{e'}\sim\mathcal{R}_{align}}[\psi(z^e(s_b^e)) - \psi(z^{e'}(s_b^{e'}))] \|_2^2 \\
&\geq \frac{1}{B}\mathbb{E}_{e,e'\sim\mathcal{E}_{train},s_b^e,s_b^{e'}\sim\mathcal{R}_{align}}[\| \psi(z^e(s_b^e)) - \psi(z^{e'}(s_b^{e'})) \|_2^2] \\
&\geq \frac{u}{B}\mathbb{E}_{e,e'\sim\mathcal{E}_{train},s_b^e,s_b^{e'}\sim\mathcal{R}_{align}}[\| z^e(s_b^e) - z^{e'}(s_b^{e'}) \|_2^2] \\
&= \frac{u}{B}\mathbb{E}_{e,e'\sim\mathcal{E}_{train},a_{0:t}^b\sim\mathcal{R}_{align},s_b^e\sim\rho(s_t^e(a_{0:t}^b)),s_b^{e'}\sim\rho(s_t^{e'}(a_{0:t}^b))}[\| z^e(s_b^e) - z^{e'}(s_b^{e'}) \|_2^2] \\
&\geq \frac{u}{2B}\mathbb{E}_{e,e'\sim\mathcal{E}_{train},a_{0:t}^b\sim\mathcal{R}_{align},s_b\sim\rho(s_t(a_{0:t}^b))}[\| z^e(s_b) - z^{e'}(s_b) \|_2^2] \qquad \text{(Lemma 4)} \\
&= \frac{u}{2B}\mathbb{E}_{e,e'\sim\mathcal{E}_{train},s_b\sim\mathcal{R}_{align}}[\| z^e(s_b) - z^{e'}(s_b) \|_2^2]
\end{aligned}
$$

This completes the proof. $\square$

## D  Additional Results

Here, we show the ablation study of PA-SF on other tasks: *Reach*, *Push*, and *Pickup*. Figure 7 demonstrates that similar results in Section 4.2 also holds on *Reach* and *Push* while on *Pickup*, PA-SF (w/o D) outperforms PA-SF on test environments marginally. We also find that the LER is relatively

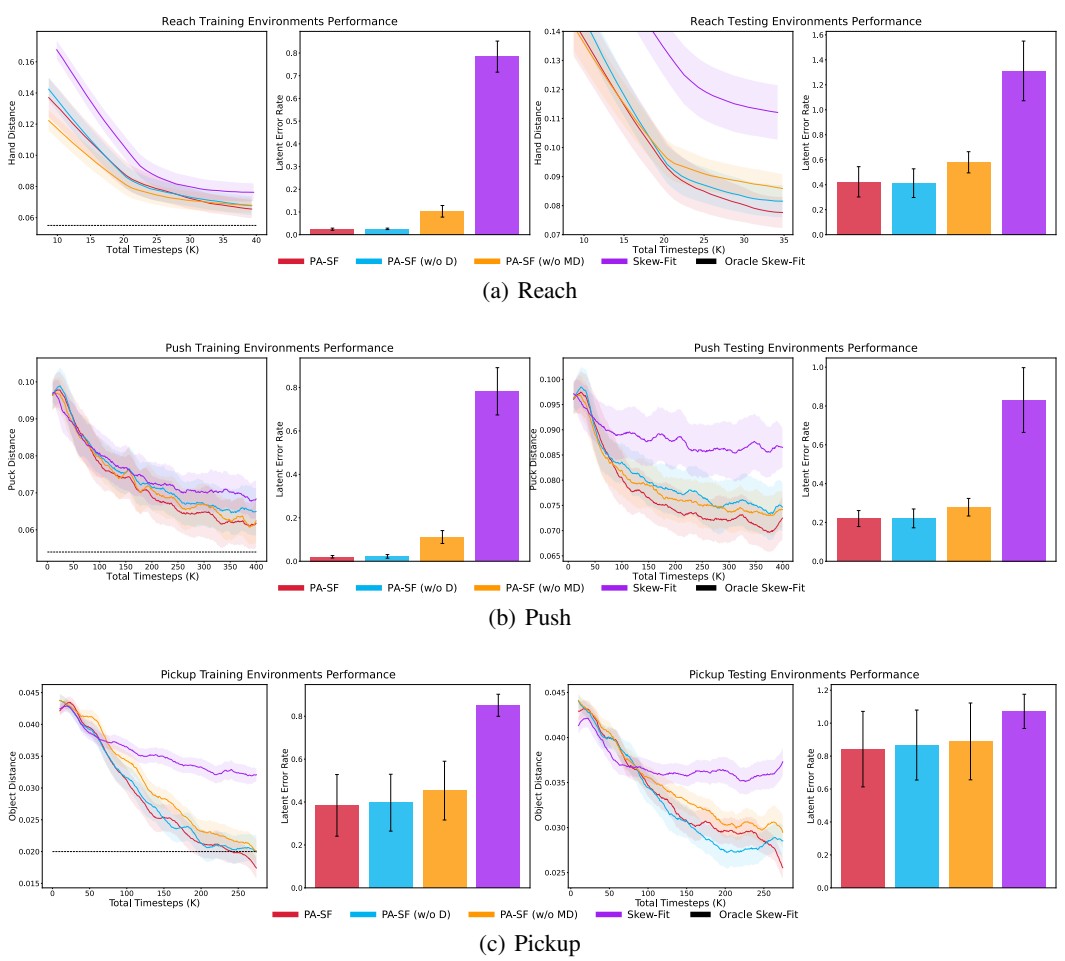

Figure 7: Ablations of our algorithm PA-SF and visualization of the latent representation via LER metric on *Reach*, *Push*, and *Pickup*. All curves represent the mean and one standard deviation (except Pickup with half standard deviation) across 7 seeds.

large in *Pickup*, perhaps owing to the relative large stochasity in this environment. However, the training and test performance are still satisfactory.

Additionally, we compare t-SNE plots of PA-SF with that of vanilla Skew-Fit in Figure 8. Clearly, the t-SNE plot generated from PA-SF are more aligned than that in Skew-Fit. Noticing that Skew-Fit also encodes the irrelevant environmental factors into the latent embedding.

Finally, we visualize how well the VAE trained with $L^{PA}$ satisfies the perfect alignment condition. In Figure 9, we visualize the reconstruction (middle line) of the original input (bottom line) as well as the shuffled reconstruction (top line), i.e., reconstruction with the same latent representation $z$ but a different environment index $e$ (Figure 2(b)). Clearly, in all tasks, the VAE successfully acquires the perfectly aligned latent space w.r.t the training environments as the shuffled reconstruction shares the almost same ground truth state $s$ with the reconstruction and the original input. Noticing that the original input images are sampled uniformly from the state space $\mathcal{S}$.

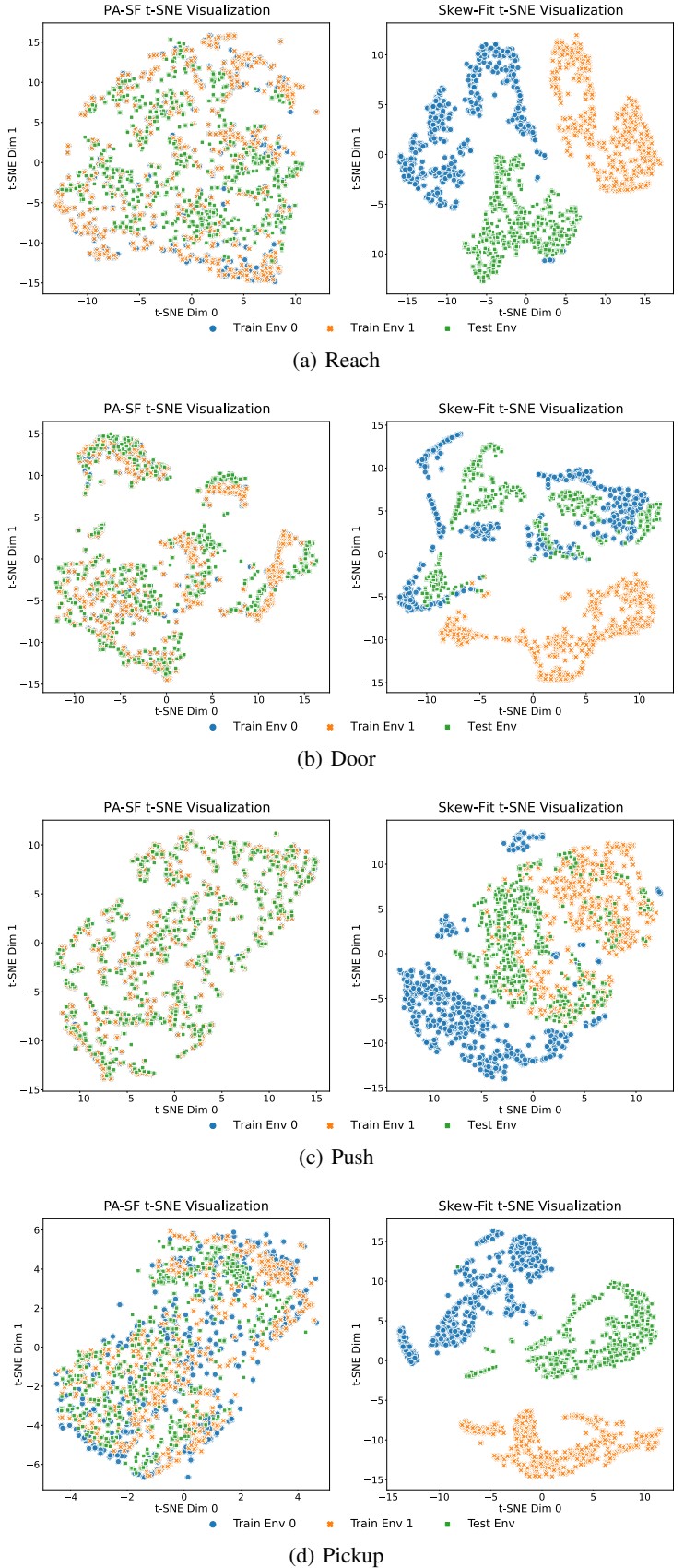

Figure 8: t-SNE visualization of the latent space $\Phi(x^e)$ trained with PA-SF and Skew-Fit for three environments, i.e., 2 training and 1 testing on different tasks.

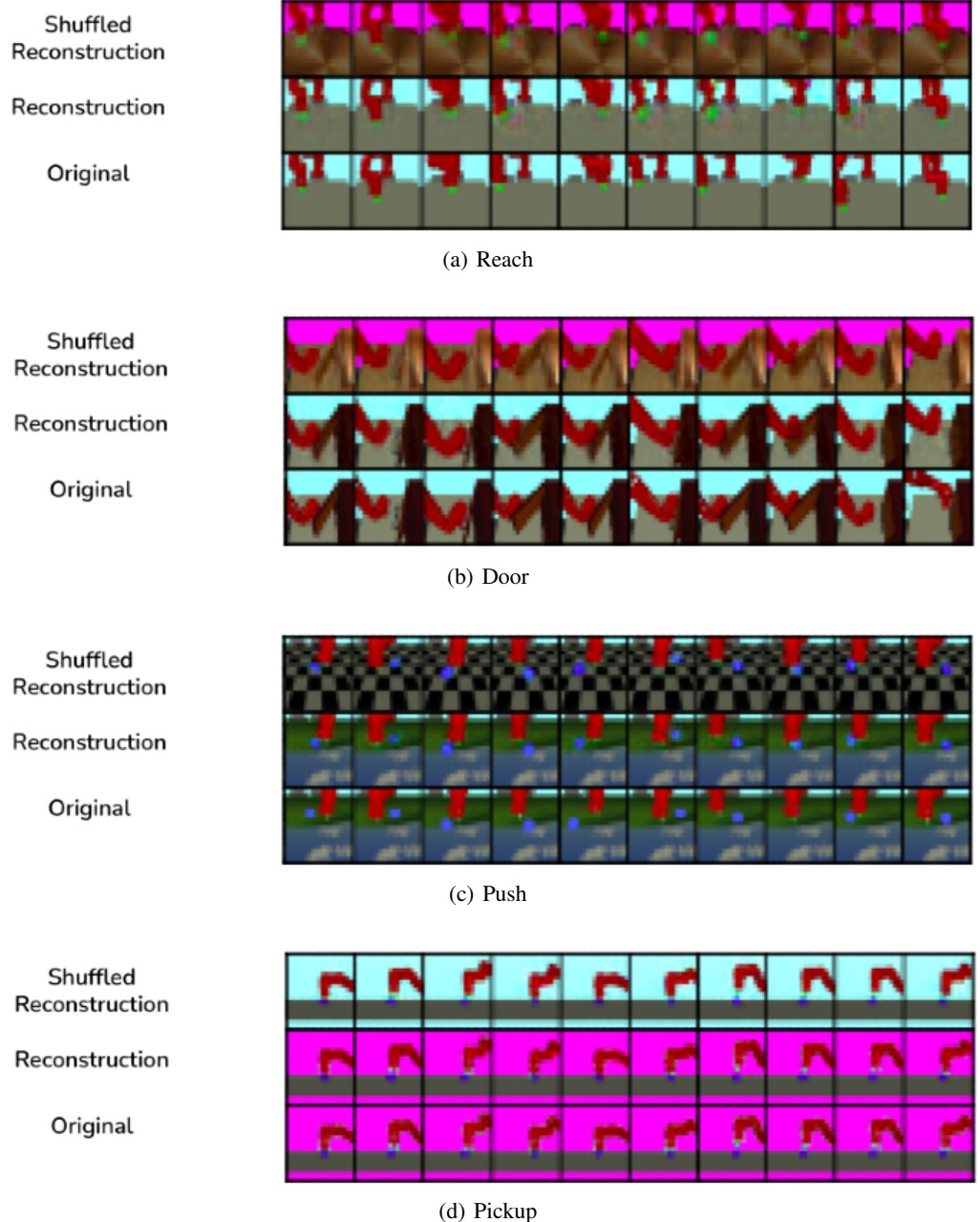

(a) Reach

(b) Door

(c) Push

(d) Pickup

Figure 9: Visualization of the VAE on all four tasks. For each figure, the bottom line shows the original input images (sampled uniformly from the state space). The middle line is the reconstruction of the input image. The top line show the shuffled reconstruction image, i.e., reconstruction with the same latent space $z$ but a shuffled environment index $e$.

# E   Experiment Details

## E.1   Task Setups

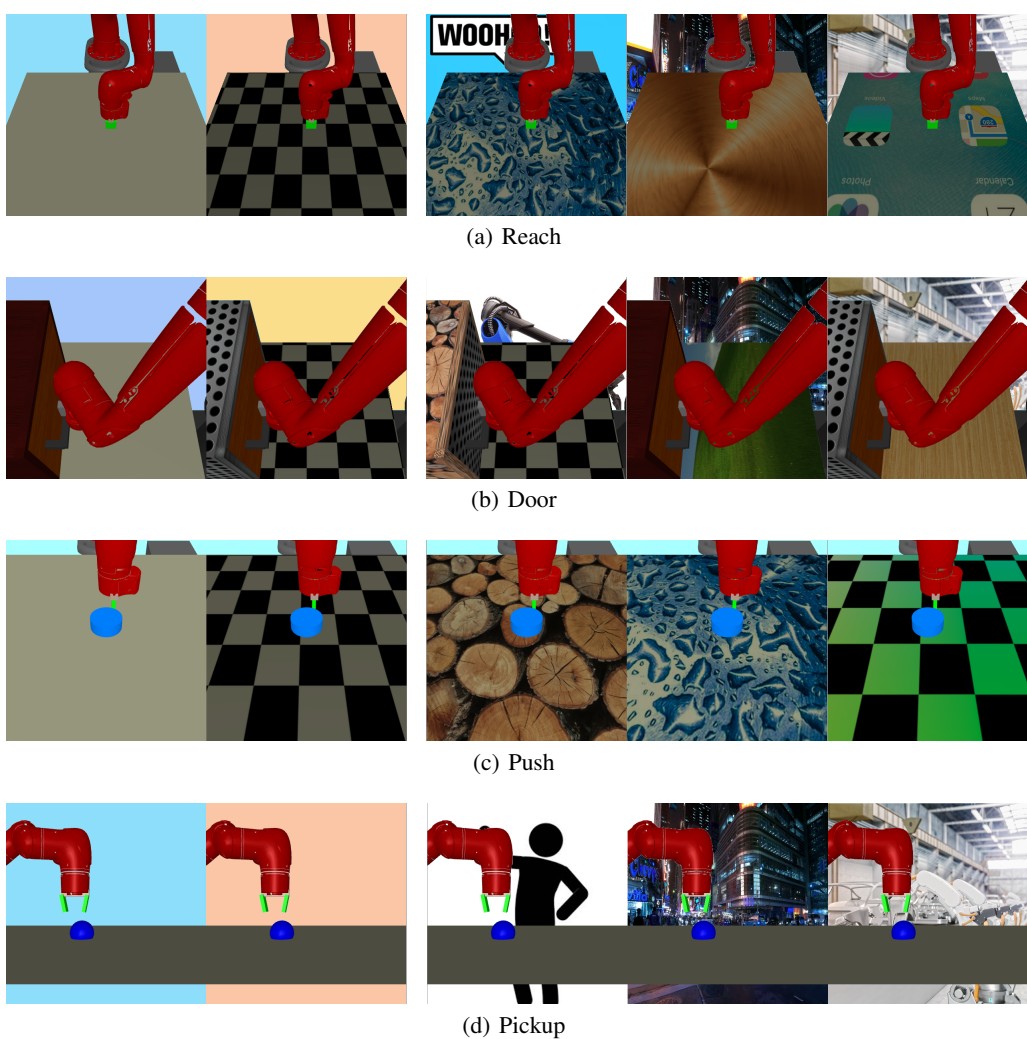

(a) Reach

(b) Door

(c) Push

(d) Pickup

Figure 10: Task setups for training (left) and test (right) environments

Our base environments are first used in Nair et al. [2018]. Figure 10 illustrates some of the environments and we provide brief descriptions as follows.

*Reach*: A 7-DoF sawyer arm task in which the goal is to reach a desired target position. We construct multiple training and test environments by altering the backgrounds with various images and dynamic videos and the foregrounds with diverse textures.

*Door*: A 7-DoF sawyer arm task with a box on the table. The goal is to open the door to a target angle. We construct multiple environments in the same way as *Reach* but with different ingredients. Additionally, we use the task with reset at the end of each episode.

*Push*: A 7-DoF sawyer arm task and a small puck on the table. The goal is to push the puck to a target position. We only change table textures as the camera has almost no background as input.

*Pickup*: The task setting is the same as *Push*. The goal is to pick up the object and place it in the desired position. We construct different environments with different backgrounds.

## E.2 Implementation Details of PA-SF

In our experiments, we use the same VAE architecture as Skew-Fit except for the environment index as an extra input for the decoder. Most of the hyper-parameters in VAE including the training schedule is the same as that in Skew-Fit except for the components we added in our algorithm. In $L^{\mathrm{MMD}}$, we use the same random expansion function as in Louizos et al. [2016] as it works well in practice. Namely,

$$\psi(z) = \sqrt{\frac{2}{D_\psi}} \cos\left(\sqrt{\frac{2}{\gamma_\psi}} W z + b\right)$$

where $z \in \mathbb{R}^d$ denotes latent embedding of observation, and $W \in \mathbb{R}^{D_\psi \times d}$ and $b \in \mathbb{R}^{D_\psi}$ are random weight and bias. $L_{\mathrm{MMD}}$ and $L_{\mathrm{DIFF}}$ are computed with respect to samples from the replay buffer and the aligned buffer. Table 3 lists the hyper-parameters that are shared across four tasks. Table 4 lists hyper-parameters specified to each task. Notice that we fine-tune the hyper-parameters on some validation environments and test on other environments.

Table 3: Shared hyper-parameters for PA-SF

| Hyper-parameter | Value |
| --- | --- |
| Aligned Path Length | 50 |
| VAE Relay Buffer Batch Size | 32 |
| VAE Aligned Buffer Batch Size | 32 |
| Random Expansion Function Dimension $D_\psi$ | 1024 |
| Random Expansion Function Scalar $\gamma_\psi$ | 1.0 |
| Number of Training per Train Loop | 1500 |
| Number of Total Exploration Steps per Epoch | 900 |

Table 4: Task specific hyper-parameters for PA-SF

| Hyper-parameter | Reach | Door | Push | Pickup |
| --- | --- | --- | --- | --- |
| MMD Coefficient $\alpha_{\mathrm{MMD}}$ | 1000 | 1000 | 200 | 100 |
| Difference Coefficient $\alpha_{\mathrm{DIFF}}$ | 0.1 | 1.0 | 0.04 | 0.04 |
| $\beta$ for $\beta$-VAE | 20 | 20 | 20 | 10 |
| Skew Coefficient $\alpha$ | -0.1 | -0.5 | -1 | -1 |
| Proportion of Aligned Sampling in Exploration | $\frac{1}{6}$ | $\frac{1}{3}$ | $\frac{1}{6}$ | $\frac{1}{6}$ |

## E.3 Implementation of Baselines

**Skew-Fit** Pong et al. [2020]: Skew-Fit is designed to learn a goal-conditioned policy by self-learning in Goal-conditioned MDPs. We extend it to Goal-conditioned Block MDPs by training the $\beta$-VAE with observations sampled from replay buffers of each training environments and constructing skewed distribution respectively. We modified some of the hyper-parameters of Skew-Fit in Table 5, did a grid search over latent dimension size, $\beta$, VAE training schedule, and number of training per train loop (Table 6).

**Skew-Fit + RAD** Laskin et al. [2020b]: Previous work Kostrikov et al. [2020] and Laskin et al. [2020b] have found that data augmentation is a simple yet powerful technique to enhance performance for visual input agents. We compare to the most well known method, i.e., Reinforcement Learning with Augmented Data (RAD) and re-implement [4] it upon Skew-Fit for Goal-conditioned Block

---

[4] https://github.com/MishaLaskin/rad

Table 5: Modified general hyper-parameters for Skew-Fit

| Hyper-parameter | Value |
|---|---|
| Exploration Noise | None |
| RL Batch Size | 1200 |
| VAE Batch Size | 96 |
| Replay Buffer Size for each $e \in \mathcal{E}_{\text{train}}$ | 50000 |

Table 6: Task specific hyper-parameters for Skew-Fit

| Hyper-parameter | Reach | Door | Push | Pickup |
|---|---|---|---|---|
| Path Length | 50 | 100 | 50 | 50 |
| $\beta$ for $\beta$-VAE | 20 | 20 | 40 | 15 |
| Latent Dimension Size | 8 | 25 | 15 | 20 |
| $\alpha$ for Skew-Fit | 0.1 | 0.5 | 1.0 | 1.0 |
| VAE Training Schedule | 2 | 1 | 2 | 1 |
| Sample Goals From | $q_\phi^G$ | $p_{\text{skewed}}$ | $p_{\text{skewed}}$ | $p_{\text{skewed}}$ |
| Number of Training per Train Loop | 1200 | 2000 | 2000 | 2000 |

MDPs. Note that RAD is originally designed to augment states for agents directly. In our setting, the augmentation is added to $\beta$-VAE training phase, which increases the robustness of latent space against irrelevant noise. Specifically, we augmented observations for training $\beta$-VAE as well as constructing skewed data distributions. At the beginning of each episode, we sample goals from the skewed distribution and encode augmented goal as latent goal. The training of SAC algorithm depends on the latent code of augmented current and next state. We also incorporate data augmentation with the skewed distribution and hindsight relabeling steps. To investigate the performance of different augmentation methods, we chose to experiment with *crop*, *cutout-color*, *color-jitter*, and *grayscale* and found that *crop* worked best among the four augmentations as reported in the RAD paper. Other augmentation methods such as *cutout-color* and *color-jitter* either replace a patch of images with single color, which may include the end-effector of Sawyer arm, or alter the color of the whole image, which may include target object (i.e., puck in *Push*) and thus hurt performance. We use the same hyper-parameters as in Skew-Fit.

**MISA** Zhang et al. [2020a] and **DBC** Zhang et al. [2020b]: Bisimulation metrics have been used to learn minimal yet sufficient representations in Block MDPs. We compare with two SOTA methods: Model-irrelevance State Abstraction (MISA) and Deep Bisimulation for Control (DBC), and modify the code for Goal-conditioned Block MDPs. In particular, we add goals' inputs into the reward predictor and use oracle ground truth distance between the current state and goal state (i.e., end-effector's position and object's position) as rewards. Our code is built upon the publicly available codes [5][6]. For fair comparison, we also fine-tuned some hyper-parameters on each task respectively. For MISA, we did a grid search over the encoder and decoder learning rates $\in \{10^{-3}, 10^{-5}\}$ and reward predictor coefficient $\in \{0.5, 1.0, 2.0\}$. For DBC, we did a grid search over the encoder and decoder learning rates $\in \{10^{-3}, 10^{-4}\}$ and bisimulation coefficients $\in \{0.25, 0.5, 1\}$. Please refer to Table 7 and Table 8 for our final choices.

Table 8: Task specific hyper-parameters for DBC

| Hyper-parameter | Reach | Door | Push | Pickup |
|---|---|---|---|---|
| Encoder and Decoder Learning Rate | $10^{-4}$ | $10^{-4}$ | $10^{-4}$ | $10^{-3}$ |
| Bisimulation Coefficient | 0.5 | 1.0 | 0.25 | 1.0 |

---

[5] https://github.com/facebookresearch/icp-block-mdp
[6] https://github.com/facebookresearch/deep_bisim4control

Table 7: Task specific hyper-parameters for MISA

| Hyper-parameter | Reach | Door | Push | Pickup |
|---|---|---|---|---|
| Encoder and Decoder Learning Rate | $10^{-5}$ | $10^{-3}$ | $10^{-3}$ | $10^{-5}$ |
| Reward Predictor Coefficient | 0.5 | 1.0 | 2.0 | 0.5 |