# OpenReview forum: "Learning Domain Invariant Representations in Goal-conditioned Block MDPs"
_NeurIPS.cc/2021/Conference — NeurIPS 2021 Poster_

### Official Review · Reviewer_KeFP · 2021-06-26

**Rating:** 6
**Confidence:** 2

**Summary:**

In the paper under review, state abstraction in Goal-conditioned RL (Block MDPs) is discussed.
Theoretical analyses to motivate the use of perfect alignment encoders are provided.
Then, a practical method (including the trajectory sampling method and encoder loss) for building the perfect alignment encoder is proposed.
Experimental results demonstrate that, in robotic arm manipulation benchmarks, the proposed method has a better generalization ability than previous methods (e.g., bi-simulation metric-based method).


**Limitations And Societal Impact:**

Yes. # Discussion about societal impact is not contained in the paper, but I think it is not really necessary for this research.

**Main Review:**

* key strengths:
1. both theoretical analyses and practical algorithm (with empirical results) are provided.
2. alignment sampling proposed in Section 3.2 is a novel idea and interesting.

* key weakness:
1 The theoretical parts (Section 3.2 and 3.3) are a bit hard to follow.
1-1.  too many symbols are used. It would be more clear if the table list of all symbols is provided.
1-2.  I am not sure if the following assumption is really valid:
1-2-1. lines 143-144: "Besides, under mild assumptions, if (E) ! 0
144 then % ! 0".
1-2-2. lines 148--149: "Though it is hard to conduct task-agnostic analysis on (t) term, we believe that perfect alignment is still an adequate criteria in minimizing (t) term".
For the assumption made at lines 148--149, I'd like to know (t) is really reduced by using a perfect alignment encoder (PA-SF) in the robotic arm control benchmarks.
2. also,   lines 145--146 (and lines 631--636 in appendix):  "Theoretically speaking, when # is a perfect alignment encoder, an goal-conditioned RL policy trained 146 over the encoded space {z(s)}s2S will minimize (E) to 0."
I think this statement is not valid if multiple optimal policies \pi_G exist. (say that there are two optimal policies \pi_{G,1} and \pi_{G,2} and that \pi is converged to \pi_{G,1}, \epsilon^{e_i}(\pi || \pi_{G, 2}) is still can be greater than zero.)

* Minor comments:
1. Typos?:
line 145:  an goal-conditioned -> a goal-conditioned
line 149:  criteria -> criterion
line 330: is formally formulated ->  is formulated
line 369: mdps -> {MDP}s

2. Reference style is not consistent (e.g., the abbreviated style of conference name is used in some references, and not in the others).

---Edit after reading the author response and the other reviews:------

I would like to improve my score, as the author has basically adequately addressed my concerns: WR->WA

I still have some concerns about the assumptions made in the theoretical analysis.
For example, I saw the author's response regarding the term (t), but was not sufficiently convinced.

I also think that the presentation of the theoretical analysis section needs to be improved (e.g., as suggested by Reviewer S2fL).



**Time Spent Reviewing:**

8

---

> ### Author Response · Authors · 2021-08-10
> **Author Response**
>
> Thank you for your valuable feedback. We address your concerns:
>
> 1. Q: Concerns on lines 143-144: "Besides, under mild assumptions, if (E) $\to$ 0 then $\delta \to 0$."
>
> A: In Appendix A.3, we provide a formal version of Proposition 2. In fact, $\delta = 0$ as long as $\Phi$ is a perfect alignment encoder. Please refer to the proof for details.
>
> 2. Q: Concerns on lines 148-149: "Though it is hard to conduct task-agnostic analysis on (t) term, we believe that perfect alignment is still an adequate criterion in minimizing (t) term."
>
> A: The term (t) contains intractable minimal and maximal operations and thus is hard to estimate directly in experiments. The discussion in lines 148-154 explains our motivation on the generalization of perfect alignment encoder and we use the empirical evaluation in lines 289-298 and Figure 4 to support the motivation. Here, we elaborate more on our logic flow and the upper bound analysis of the term (t).
>
> In Appendix A.3, we discuss the (t) term on the condition that the encoder is a perfect alignment over both training and testing environments. Ideally, the (t) term can be minimized given this condition. The analysis can be further extended to an error-bound analysis, e.g., if there exists $\eta$ probability of states in $\rho^t_{\pi_G}$ that are not perfectly aligned, we can bound that $(t) \leq O(\eta + \delta)$. The result suggests that if the more aligned the encoder is, the smaller the upper bound is and so as the (t) term to some extent. In the empirical evaluation (Figure 4), the LER metric suggests that via $L^{\text{PA}}$, we acquire an encoder that is more aligned over the testing environments than not using the loss. There is also a clear correspondence between the scale of LER metric and the performance metric over testing environments (Figure 4). This suggests a strong correlation between the quality of the latent representation and the generalization performance. To sum up, based on the (t) term upper bound and the empirical evidence, we believe that the (t) term is reduced via the proposed method and so as the actual performance $J^t(\pi)$ in the robot arm benchmark.
>
> We have to admit that analyzing the (t) term directly is inherently difficult and there may exist certain $\rho^t(x, g)$ that has a large (t) value. From a theoretical perspective, this suggests that given limited samples and training environments, the agent's o.o.d generalization performance can only be bounded within a small range away from the training environments, i.e., the characteristic set which is larger than the interpolation convex hull. Indeed, such theoretical limitation is also notified in the domain generalization literature [Sicilia et al., 2021].
>
> 3. Q: Concerns on lines 145-146 (and lines 631--636 in appendix): "Theoretically speaking, when $\Phi$ is a perfect alignment encoder, an goal-conditioned RL policy trained over the encoded space $\{z(s)\}$s will minimize $(E)$ to $0$."
>
> A: Proposition 1 and Proposition 2 hold for any policy $\pi$ and any optimal policy $\pi_G$. For tasks where there exist multiple optimal policies, the RL policy $\pi$ trained over a perfect alignment encoder will converge to one of the optimal policies $\pi_G^1$ and thus the $(E)$ term will achieve $0$ w.r.t $\pi_G^1$. Thus, in this statement (lines 145-146), we refer to the optimal policy in Proposition 2's bound as $\pi_G^1$ correspondingly. We will clarify the ambiguity in the next version.
>
> [1] Sicilia, A., Zhao, X., \& Hwang, S. J. (2021). Domain adversarial neural networks for domain generalization: When it works and how to improve. arXiv preprint arXiv:2102.03924.

---

> > ### Author Response · Authors · 2021-08-10
> > **List of Symbols**
> >
> >  | Symbol                                                       | Description                                                  |
> >  | ------------------------------------------------------------ | ------------------------------------------------------------ |
> >  | $M^{\mathcal{E}}$                                            | Goal-conditioned Block MDPs                      |
> >  | $e$                                                          | Environment index                                            |
> >  | $\mathcal{S}$                                                | Shared state space among environments $e \in \mathcal{E}$    |
> >  | $\mathcal{A}$                                                | Shared action space among environments $e \in \mathcal{E}$   |
> >  | $\mathcal{X}^e$                                              | Specific observation space for environment $e$               |
> >  | $\mathcal{T}^e$                                              | Specific dynamic transition for environment $e$              |
> >  | $\mathcal{G}$                                                | Shared goal space among environments $e \in \mathcal{E}$     |
> >  | $\gamma$                                                     | Discount factor  |
> >  | $b^e$                                                        | Environmental factor for environment $e$ (e.g. background)   |
> >  | $\mathcal{B}^e$                                              | Specific environmental factor space for environment $e$ (i.e. video backgrounds are allowed) |
> >  | $x^e_t = x^e(s_t, b^e_t)$                                    | Observation determined by state $s$ and environmental factor $b^e$ for environment $e$ at timestep $t$ |
> >  | $p(s_{t + 1} \vert s_t, a_t)$                                | State transition shared among environments                   |
> >  | $q^e(b^e_{t + 1} \vert b^e_t)$                               | Environmental factor transition for environment $e$          |
> >  | $\mathcal{X} = \cup_{e \in \mathcal{E}} \mathcal{X}^e$       | Joint set of observation spaces                              |
> >  | $\pi(a \vert x^e, g)$                                        | Goal-conditioned policy            |
> >  | $J(\pi)$                                                     | Objective function for policy $\pi$                          |
> >  | $J^e(\pi)$                                                   | Objective function for policy $\pi$ in environment $e$       |
> >  | $p_{\pi}^e(s_t=g \vert g)$                                   | Probability of achieving $g$ under policy $\pi$ at timestep $t$ in environment $e$ |
> >  | $\pi_G(\cdot \vert x^e, g)$                                  | One of the optimal policies which are invariant over all environments |
> >  | $\\{e_i\\}_{i=1}^N$              | Training environments                                        |
> >  | $\rho(x, g) = \rho(g) \rho(x \vert g)$                       | Joint distributions of goals and observations                |
> >  | $\rho_{\pi}^e(x^e \vert g)$                                  | Occupancy measure of $x^e$ in environment $e$ under policy $\pi$ |
> >  | $\rho_{\pi}^e(x^e)$                                          | Marginal distribution of $\rho_{\pi}^e(x^e, g)$ over goals |
> >  | $\epsilon^{\rho(x,g)}(\pi_1 \parallel \pi_2)$                | Averaged Total Variation between policy $\pi_1$ and $\pi_2$  |
> >  | $\Pi$                                                        | Policy class (i.e. space for all possible policies)          |
> >  | $d_{\Pi \Delta \Pi}(\rho(x, g), \rho(x, g)')$                | $\Pi \Delta \Pi$-divergence of two joint distributions $\rho(x, g)$ and $\rho(x, g)'$ in terms of the policy class $\Pi$ |
> >  | $\epsilon^{e_i}(\pi_1 \parallel \pi_2) = \epsilon^{\rho_{\pi}^{e_i}(x^{e_i}, g)}(\pi_1 \parallel \pi_2)$ | Total Variation between policy $\pi_1$ and $\pi_2$ averaged over joint occupancy measure under policy $\pi$ in training environment $e_i$ |
> >  | $\epsilon^t(\pi_1 \parallel \pi_2) = \epsilon^{\rho_{\pi_G}^t(x^t, g)}(\pi_1 \parallel \pi_2)$ | Total Variation between policy $\pi_1$ and $\pi_2$ averaged over joint occupancy measure under policy $\pi_G$ in testing environment $t$ |
> >  | $\pi^*$                                                      | The closest policy for training environments in policy class $\Pi$ w.r.t. optimal invariant policy $\pi_G$ measured by averaged Total Variation |
> >  | $\delta$                                                     | Maximum $\Pi \Delta \Pi$-divergence between occupancy measure for two different training environments under given policy $\pi$ |
> >  | $\lambda$                                                    | Performance of $\pi^*$ in both training and testing environments in terms of averaged Total Variation |
> >  | $B$                                                          | Characteristic set of joint distributions determined by $\mathcal{E}_{\text{train}}$ and policy class $\Pi$ |
> >  | $\Phi$                                                       | Observation encoder                                          |
> >  | $z^e_t(s) = \Phi(x^e_t(s_t))$                                | Latent representation of observation $x^e_t$ with state $s_t$ in environment $e$ |
> >  | $\Pi_{\Phi} = \\{w \circ (\Phi(x), g), \forall w\\}$           | Policy class induced by encoder $\Phi$ with any function $w$ |
> >  | $\tilde{\rho}(x, g)$                                         | The closest occupancy measure in characteristic set $B$ w.r.t. occupancy measure in testing environment under $\Pi \Delta \Pi$-divergence |
> >  | $\\{s^e_t(a_{0:t})\\}_{t=0}^T$                                 | Set of states along trajectory $\{x_0^e, a_0, x_1^e, a_1, \ldots, x^e_T\}$ |
> >  | $\\{x_t^{e_i}(a_{0:t})\\}_{t=0}^T$                                     | Aligned observations for environment $e_i$                   |
> >  | $\mathcal{R}_{\text{align}}$                                 | Replay buffer for aligned transitions                        |
> >  | $\mathcal{B}_{\text{align}}$ | Batch of aligned observations   |
> >  | $L^{\text{MMD}}(\Phi)$                                       | MMD loss for encoder $\Phi$                                  |
> >  | $\psi(z)$                                                    | Random expansion function for latent representation $z$      |
> >  | $L^{\text{DIFF}}(\Phi)$                                      | Difference loss for encoder $\Phi$                           |
> >  | $\mathcal{R}^e$                                              | Replay buffer for transitions from environment $e$           |
> >  | $L^{\text{PA}}$                                              | Perfect alignement loss                                      |
> >  | $L^{\text{RECON}}$                                           | Reconstruction loss                                          |
> >  | $\beta$                                                      | KL divergence coefficient                                    |
> >  | $\alpha_{\text{MMD}}$                                        | MMD loss coefficient                                         |
> >  | $\alpha_{\text{DIFF}}$                                       | Difference loss coefficient                                  |
> >  | $\text{Err}(\Phi)$                                           | Latent error rate for encoder $\Phi$                         |

---

### Official Review · Reviewer_6X8Q · 2021-07-14

**Rating:** 6
**Confidence:** 3

**Summary:**

This paper aims to learn a domain invariant representation of observations in the goal-conditioned block MDP (GBMDP). To this end, the authors propose to minimize the deviation between embedded visitation trajectories with a fixed action sequence across different environments. The intention is to learn the latent mapping between states and observations. The authors also characterize the optimality gap between an arbitrary policy and the optimal invariant policy. Their analysis shows that the key to optimal and invariant policy is (i) the optimality of the policy at each environment, (ii) the deviation of the visitation measures across different environments, and (iii) the difference between the test environment and training environments.  The authors conduct extensive experiments on the multi-world environment and show that the proposed PA-SF method outperforms several baselines.

**Limitations And Societal Impact:**

See Limitations and inquiries in the Main review. I do not have concerns regarding the societal impact of this work.

**Main Review:**

Strength:

1. Learning invariant representation that can generalize across different environments is an important challenge in the study of RL. The authors propose a novel algorithm that tackles such a challenge and outperforms several strong baselines on the multi-world environment.

2. The authors propose an analysis of the optimality gap in  GBMDP and identify several important factors of being optimal in solving GBMDP. The authors further propose their algorithm based on their theoretical findings.

Limitations and inquiries

1. In Proposition 1, is the bound tight? It seems that $\delta$ measures the discrepancy of observation visitations across different environments, which can hardly go to zero due to different observations generated by the corresponding environments. Under what scenario will the term $\delta$ become zero?


2. In the aligned sampling part, it seems to me that the visitation also depends on the initial states. Hence, for different visitation measures to attain the same latent state distribution under the same action sequence, one should also ensure that the initial state distributions are the same. I wonder how to ensure that the initial observation distributions share the same latent state across different environments?


3. Despite the fact that the test environment in this paper is widely accepted, I wonder if swapping only the background grant sufficient challenge in learning to transfer. In fact, with only changes in the background, it is hard to tell if the performance is due to the trained VAE that filters out the background or if the agent learns a correspondence between the pixel input and a latent state. For instance, what happens if one changes the locations of the camera in the training and test environments? Will the agent learn the latent state corresponding to the position of the robot arm in such a scenario?

**Time Spent Reviewing:**

3

---

> ### Author Response · Authors · 2021-08-10
> **Author Response**
>
> Thank you for your valuable feedback. We address your concerns:
>
> 1. Q: Under what scenario will the term $\delta$ become zero?
>
> A: As shown in Proposition 2 (more formally stated in Appendix A.3), $\delta \to 0$ when $\Phi$ becomes a perfect alignment encoder over the training environments. Empirically, we find that the encoder does satisfy the perfect alignment property upon convergence (section 4.2).
>
> 2. Q: Assumptions on the initial state distributions.
>
> A: We do assume that all environments share the same state transition dynamics including the initial state distribution $p(s_0)$. Environments are reset after each trajectory ends. This formulation is used in many DRL algorithms, e.g., SAC, PPO. Most benchmarks also use the same formulation, e.g., multiworld [Nair et al., 2018] or standard OpenAI Gym [Brockman et al., 2016]. In the robot arm application, this requirement can be easily satisfied, as we can simply reset the robot arm to a default position with small random noise.
>
> 3. Q: Interpretation of the experiments and beyond.
>
> A: In our environment setup, a VAE that filters out the background also learns the correct correspondence between the pixels and the latent representation (the correct correspondence between the arm, the object and the latent representation $z$ which should be invariant of background changes) and an encoder that learns the correct correspondence also indicates that the VAE filters out the background information. In the experiment, we have demonstrated that this correspondence is acquired via the $L^{\text{PA}}$ loss.
>
> As you suggested, we also conduct the multiple camera views experiment to test the applicability of our proposed method. In this experiment, we use the same background but differ the camera view in different environments. Thus, an invariant policy has to capture the correct spatial information of the underlying state. The result is shown in the following table. Clearly, ours still perform well in this experiment while other methods fail to generalize to unseen test environments.
>
> |                    | Reach                      | Door                       | Push                       |
> | ------------------ | -------------------------- | -------------------------- | -------------------------- |
> | Algorithm          | Hand distance (35K)        | Angle difference (120K)    | Puck distance (400K)       |
> | **Skew-Fit**       | $0.117 \pm 0.011$          | $0.258 \pm 0.031$          | $0.096 \pm 0.007$          |
> | **Skew-Fit + RAD** | $0.112 \pm 0.007$          | $0.254 \pm 0.023$          | $0.096 \pm 0.006$          |
> | **PA-SF**          | $\mathbf{0.074} \pm 0.009$ | $\mathbf{0.180} \pm 0.037$ | $\mathbf{0.081} \pm 0.006$ |
>
> [1] Brockman, G., Cheung, V., Pettersson, L., Schneider, J., Schulman, J., Tang, J. and Zaremba, W., 2016. Openai gym. arXiv preprint arXiv:1606.01540.
>
> [2] Nair, A., Pong, V., Dalal, M., Bahl, S., Lin, S., \& Levine, S. (2018). Visual reinforcement learning with imagined goals. arXiv preprint arXiv:1807.04742.

---

> > ### Comment · Reviewer_6X8Q · 2021-09-02
> > **Re: Author Response**
> >
> > Thanks for the response and the additional experimentation. The extra experiments on environments with different views are convincing to me. The paper could be further strengthened if the authors could come up with other transfer learning scenarios (other than view change and background change). Also, the theory could be stronger if the authors discuss what happens if imperfect alignment is adopted. Overall, my concerns are addressed adequately and I have raised my score to 6.

---

### Official Review · Reviewer_WvNf · 2021-07-18

**Rating:** 6
**Confidence:** 3

**Summary:**

Summary
-------

This paper proposes a pair of regularizers to the Skew-Fit algorithm to
ensure that the encoder maps two states to the same encoding if and only
if the states are the same (whether the states are in different
environments or not). The problem setting is goal-conditioned RL but the
paper introduces the block MDP structure to better analyze
generalization. The perfect alignment property is motivated from a
derived inequality measuring the performance difference of two policies
with ideas based on the $\mathcal{H}\Delta \mathcal{H}$ divergence. The
proposed algorithm is found to be most performant on multiworld amongst
the baselines: SkewFit, MISA and DBC. The performance benefit is
attributed to the enforcement of the perfect alignment property.

.


**Limitations And Societal Impact:**

The authors adequately addressed the limitations and potential negative societal impact.

**Main Review:**

Decision
--------

The technical novelty and reported performance benefit of the algorithm
seems very compelling. I have, however, some reservations about the
perfect alignment property and the way it is enforced through the
regularizer. Based on these reservations and some issues with the
empirical investigation, I am rating the paper at a 5 ("Marginally below
the acceptance threshold"). If, after discussing with the authors and
other reviewers, I am able to clear my reservations then I will be
willing to increase the score marginally. Below, I have outlined my
concerns more specifically and I hope it will help the authors improve
their paper further.

Strengths
---------

-   The theoretical analysis proposed in this paper seems novel. In
    particular, the combination of goal-conditioned RL and block MDPs,
    and the resultant domain generalization theory, is interesting.
    Although the proof of lemma 1 seems similar to largely follow the
    technical machinery in most theoretical RL papers, proposition 1 is
    still insightful. Perfect alignment is somewhat motivated from this
    bound, but this reasoning is loose due to the challenging (t) term
    in Equation 3.
-   Overall problem is well described and motivated.
-   The empirical results on the mulitworld environment suggest that the
    proposed algorithm does benefit from improved generalization. The
    choice in baselines seems comprehensive, and the ablation study does
    demonstrate the importance of the particular combination of
    techniques for perfectly aligned Skew-Fit.

Weaknesses
----------

-   Perfect alignment seems at odds with fundamental properties of the
    reinforcement learning problem. In a particular environment $e$, two
    states $s$ and $s'$ could have the same policy
    $\pi(a | s) = \pi( a | s')$ or, given an arbitrary policy, the two
    states could have equal value $\pi$, $V^\pi(s) = V^\pi(s')$. If the
    encoder is perfectly aligned, and we use a linear predictor $w$ on
    top of the encoded state $z$, then I do not think you could
    represent equivalences of value/policy between different states.
-   Empirical concerns: I understand that using a self-supervised
    goal-conditioned RL formulation requires a certain environment setup
    (i.e. goal formulation and semantic observations). I think this does
    not disqualify some classic control environments like mountaincar,
    which would help contextualize your results.
-   Ablation on regularization hyperparameters. Your main novelty is the
    introduction of a regularizer to enforce perfect alignment. With the
    addition of these regularizers however, you also introduce more
    degrees of freedom in the form of the hyperparameters controlling
    the regularization strength. I think there should be an ablation
    study for these hyperparameters to ensure that the performance gains
    is not attributed to the extra degrees of freedom instead of the
    property of perfect alignment.

Detailed Comments
-----------------

-   Importance of aligned sampling vs observation sampling is unclear.
    The discussion in Section 3 is brief and the importance of splitting
    the data collection to only partially aligned sampling is not
    investigated in the experiments.

Minor Comments
--------------

-   Line 71: The disjoint property suggests that the environment $b^e$
    is not actually necessary in constructing the observation. What is
    the role of $b^e$ in this case?
-   What is Figure 5 supposed to show? There is no clustering of the
    points but I am unsure if that is to be expected. It seems that the
    puck position may be the same on left and the same on the right.
    This could be better visualized through a plot of latent distance
    and puck distance

Post Rebuttal
--------------

In light of the discussion below, I have increased my score from a 5 -> 6.

**Time Spent Reviewing:**

6

---

> ### Author Response · Authors · 2021-08-10
> **Author Response**
>
> Thank you for your valuable feedback. We address your concerns:
>
> 1. Q: Sufficient representation power of the policy class and the critic.
>
> A: We use policy class $\Pi_{\Phi} = \{w \circ \Phi, \text{for all function } w\}$ and we implement $w$ as a neural network in the experiment instead of a linear predictor. Therefore, the representation power of the policy class and the critic is large enough to represent the equivalences of value/policy between different states.
>
> 2. Q: Ablation on regularizer's hyperparameters.
>
> A: We conducted an ablation study across all four environments, varying $\alpha_{\text{MMD}} \in \{0.01, 0.1, 1.0\}$ and $\alpha_{\text{DIFF}} \in \{10, 100, 1000\}$. Our algorithm PA-SF outperforms SkewFit in 17 of 18 these configurations on Reach and Door environments. This suggests the robustness of PA-SF against hyperparameter's choices. Moreover, we also find that learning with larger $\alpha_{\text{MMD}}$ coefficient often results in better generalization performance in testing environments. This supports our claim that the more aligned the encoder is, the better the generalization is.
>
> Table 1: Hyperparameter ablation of PASF on Testing Environments of Reach
>
> |                        |                              |                   | $\alpha_{\text{MMD}}$ |                   |                   |
> | ---------------------- | ---------------------------- | ----------------- | --------------------- | ----------------- | ----------------- |
> |                        | Test Avg Hand distance (35K) | 10.0              | 100.0                 | 1000.0            | SkewFit           |
> |                        | 0.01                         | $0.080 \pm 0.011$ | $0.086 \pm 0.007$     | $0.079 \pm 0.015$ |                   |
> | $\alpha_{\text{DIFF}}$ | 0.1                          | $0.125 \pm 0.061$ | $0.088 \pm 0.010$     | $0.092 \pm 0.027$ | $0.112 \pm 0.019$ |
> |                        | 1.0                          | $0.086 \pm 0.010$ | $0.086 \pm 0.010$     | $0.084 \pm 0.012$ |                   |
>
>
>
> Table 2: Hyperparameter ablation of PASF on Testing Environments of Door
>
>
>
>
> |                        |                                  |                   | $\alpha_{\text{MMD}}$ |                   |                   |
> | ---------------------- | -------------------------------- | ----------------- | --------------------- | ----------------- | ----------------- |
> |                        | Test Avg Angle difference (150K) | 10.0              | 100.0                 | 1000.0            | SkewFit           |
> |                        | 0.01                             | $0.156 \pm 0.056$ | $0.127 \pm 0.045$     | $0.095 \pm 0.032$ |                   |
> | $\alpha_{\text{DIFF}}$ | 0.1                              | $0.131 \pm 0.026$ | $0.097 \pm 0.023$     | $0.090 \pm 0.025$ | $0.162 \pm 0.036$ |
> |                        | 1.0                              | $0.146 \pm 0.043$ | $0.155 \pm 0.057$     | $0.118 \pm 0.049$ |                   |
>
>
> 3. Q: Importance of aligned sampling v.s. observation-dependent sampling.
>
> A: Aligned sampling is important as it provides aligned data for learning perfect alignment encoder via $L^{\text{PA}}$ while observation-dependent sampling cannot. In section 4.2, we conduct the ablation study of PA-SF without aligned sampling (PA-SF w/o AS). Clearly, to deliberately align unaligned data deprecates the latent representations and the performance drops significantly on both training and testing environments (Figure 4). In addition, learning without $L^{\text{PA}}$ and aligned sampling also has worse performance (PA-SF w/o MD, SkewFit). These results suggest that aligned sampling is important for perfect alignment and observation-dependent sampling can not achieve the same result.
>
> However, only using aligned sampling for exploration reduces sample efficiency. Aligned sampling results in $N = |\mathcal{E}_{\text{train}}|$ times samples for exploration as it repeats the same sequence of actions in all training environments. In our experiment, aligned sampling only accounts for 15\% of all exploration samples. In this way, PA-SF can solve all tasks as quickly as were trained in a single environment (Oracle Skew-Fit, Figure 4).
>
> 4. Q: What is the role of $b^e$ ?
>
> A: We use the variable $b^e$ to emphasize the different environmental factors of different environments $e \in \mathcal{E}$ of GBMDP problems. The formulation follows prior work in [Zhang et al., 2020].
>
> 5. Q: What is Figure 5 supposed to show?
>
> A: The t-SNE plot visualizes the representation space, i.e., $z(x^e(s))$ for different $e \in \mathcal{E}$ of the same oracle state dataset. The plot shows qualitatively how well the perfect alignment is learned via our method. Orange and blue dots are samples from two training environments and they are almost perfectly overlapped, i.e., aligned in the latent space. The dots of the testing environment is near the corresponding training dots (of the same underlying state $s$ as the illustrative images on the left and right). This shows that a perfect alignment encoder over training environments also increases the alignment over testing environments (though not as perfect as the training environments). The claim is also supported by the LER metric in Figure 4.
>
> [1] Zhang, A., Lyle, C., Sodhani, S., Filos, A., Kwiatkowska, M., Pineau, J., ... \& Precup, D. (2020, November). Invariant causal prediction for block MDPs. In International Conference on Machine Learning (pp. 11214-11224). PMLR.

---

> > ### Comment · Reviewer_WvNf · 2021-08-19
> > **Thank you for clarifying**
> >
> > Thank you for clarifying, especially with respect to using only aligned sampling and the additional ablation experiments. The first point doesn't answer my query exactly.
> >
> > I don't think the issue is with representation power of the policy/critic. Rather, that the perfectly aligned property is itself not always desirable. Consider two states $s, s'$ in environment $e$ that are distinct $s \not = s'$, but have the same value $V(s) = V(s')$. A perfectly aligned encoder will learn distinct encodings $\phi(x^e(s)) \not = \phi(x^e(s'))$ that do not help, pushing the problem to the outer neural network $w$. Now consider a single state $s$ in environment $e, e'$, where $V_e^*(s) \not = V_{e'}^*(s)$ but $\phi(x^e(s)) = \phi(x^{e'}(s))$. In this case, a perfectly aligned encoder would also not be helpful, and the problem is again pushed to the outer neural network $w$. The perfect alignment property seems limited to environments where the task similarity is similar enough that the values/policies are basically the same between different environments, and that different states in a single environment are quite different.

---

> > > ### Author Response · Authors · 2021-08-22
> > > **Validity of Perfect Alignment**
> > >
> > > Thank you for raising the concern--we are happy to clarify further. From a theoretical perspective, we believe the perfect alignment condition is a desirable property as it reduces the RHS of Eq. (2) to Eq. (3) which is smaller. We address the two scenarios you mentioned as follows.
> > >
> > > (1) $\Phi(x^e(s)) \neq \Phi(x^e(s'))$ with $V^{\pi}(s) = V^{\pi}(s')$.
> > >
> > > In goal-conditioned problems, the value function takes both current state and goal state as inputs, i.e., $V^{\pi}(s, g)$. In GBMDP, we choose $V^{\pi}_e(s, g) = v \circ (\Phi(x^e(s)), g)$. Though $V^{\pi}_e(s, g_0)$ may equal to $V^{\pi}_e(s', g_0)$ and $s \neq s'$ for some goal $g_0$, there usually exists many other goals $g$ that $V^{\pi}_e(s, g) \neq V^{\pi}_e(s', g)$. Thus, $\Phi(x^e(s))$ should be differ from $\Phi(x^e(s'))$ otherwise the value estimation will be inaccurate for the other goals.
> > >
> > > For example, we consider the value function of the optimal policy in the puck pushing task. For any two states $s, s'$ (puck position), it has the same $V^*_e$ when the goal position is the middle point of the line between the two states. However, for all the other goals on the line, $V^*_e(s, g) \neq V^*_e(s', g)$. Clearly, it is generally essential that $\Phi(x^e(s)) \neq \Phi(x^e(s'))$ if $s \neq s'$.
> > >
> > > (2) $\Phi(x^e(s)) = \Phi(x^{e'}(s))$ while $V_e^*(s) \neq V_{e'}^*(s)$.
> > >
> > > In GBMDP (section 2), all environments share the same transition $p(s'|s, a)$ and only differ in the environmental factor $b^e$ which is irrelevant with the goal-conditioned problem (since $\mathcal{G} \subset \mathcal{S}$). Consequently, for any $g$ and environments $e, e'$, $V_e^*(s, g) = V_{e'}^*(s, g)$ for the optimal policy. In addition, for any policy $\pi = w \circ (\Phi(x^e(s)), g)$ where $\Phi$ is a perfect alignment encoder (i.e., the policy is invariant of the environmental factors), it always has $V_{e}^{\pi}(s, g) = V_{e'}^{\pi}(s, g), \forall s, g$.Thus, it is consistent to have $\Phi(x^e(s)) = \Phi(x^{e'}(s))$ and to estimate the correct value.

---

> > > > ### Comment · Reviewer_WvNf · 2021-08-31
> > > > **Re: Validity of Perfect Alignment**
> > > >
> > > > Thank you for the futher clarification, especially since my notation did not use the goal factorization $V_e^\pi(s,g) = v \circ (\Phi(x^e(s)), g)$.
> > > >
> > > > For the first point, you state that there exists other goal states where having the perfect alignment property is valid. The question comes down to whether or not the joint set of (g,s,s') in agreement ($V^\pi(s,g) = V^\pi(s',g)$) is larger than the set of (g,s,s') in disagreement. I don't doubt that the set in disagreement is larger, but these assumptions/limitations underlying the perfect alignment property should be presented upfront.
> > > >
> > > > On the second point, I remain a bit confused. On lines 242-244, you mention that you also include videos that would alter the underlying environment factor transition. Wouldn't this prevent value equality for an arbitrary policy?

---

> > > > > ### Author Response · Authors · 2021-08-31
> > > > > **Further Clarification**
> > > > >
> > > > > Thanks for your feedback. We address your concerns:
> > > > >
> > > > > For the first case, as long as there exists $g$ that $V_e^{\pi}(s, g) \neq V_e^{\pi}(s', g)$, we must have $\Phi(x^e(s)) \neq \Phi(x^e(s'))$, otherwise we cannot approximate the values for goal $g$ correctly. Therefore, in theory, even when the set in disagreement (non-empty) is smaller than the set in agreement, we still need separated latent embeddings of $s, s'$. The perfect alignment condition is a general criterion that will not cause the inapproximability problem of value functions. Thus, from a theoretical perspective, the encoder should satisfy the *only if* condition for general GBMDPs. We will discuss more on this in our next version.
> > > > >
> > > > > For the second case, video background environments represent that the environmental transition $q(b^e_{t+1}|b^e_t)$ is non-identity. As illustrated in Figure 1, in GBMDPs, changing environmental factors will not affect the underlying MDP's dynamic $p(s'|s, a)$. For example, a man walking through the background will not affect the task of the robot in the foreground.
> > > > >
> > > > > Therefore, for any policy $\pi \in \Pi_{\Phi}$ whose $\Phi$ is a perfect alignment, it still satisfies that $V_e^{\pi}(s, g) = V_{e'}^{\pi}(s, g)$ with changing environmental factors. Noticing that the equality property may not hold for policies beyond $\Pi_{\Phi}$ ($\Phi$ is a perfect alignment). Thus, we aim to acquire a perfect alignment encoder $\Phi$ in the very first place for both the policy and the value function.

---

### Official Review · Reviewer_S2fL · 2021-07-18

**Rating:** 6
**Confidence:** 4

**Summary:**

**High-Level Summary**

This paper studies a new representation learning approach for goal-conditioned RL for the particular case of Block MDPs (Du et al., 2019). The goal is to learn a goal-conditioned policy that generalizes across a set of Block MDPs given access to a train set of Block MDPs. It is assumed that latent dynamics are nearly deterministic which is critical. Formally, the proposed representation learning approach samples sequence of actions and executes them in different sampled environments. An encoder is trained to map observations at the same time step along these two trajectories to a similar value (line 173) while separating two randomly chosen observations (line 186). It is argued that learning this representation yields invariant representation which helps optimize the generalization error. Experimental evidence supports the claim.

**Limitations And Societal Impact:**

See main review for limitations.

**Main Review:**

**Technical Summary**

This paper studies goal-conditioned RL in a set of environments where the goal is to learn a goal-conditioned policy that reaches the goal in different environments. The setting assumes that these environments share a latent state space, action space, latent transition dynamics, and goal states which are a subset of state space. Further, a set of environment factors specific to each environment evolve independently of actions and are combined with the current latent state to generate an environment-specific observation. It is assumed that the latent state transition dynamics is nearly deterministic though the exact assumption is not formally stated in the paper. It can be shown that an optimal policy $\pi_G$ exists that reaches the goal optimally and only depends on the latent state.

The first set of results analyze the generalization performance of a policy $\pi$ compared to $\pi_G$. This is similar to domain-generalization results. A set of policy is considered that factorizes as $\pi(x) = \pi(\phi(x))$. It is argued that if $\phi$ is a perfectly aligned decoder that maps two observations to the same value only if they are from the same state, then a policy trained on top of it will minimize the generalization error. This is not mathematically proven in the paper. Finally, the learned decoder is combined with the skew-fit algorithm that trains a soft actor-critic on a reward function that is given by the distance between the encoding of current state and goal and additionally trains the encoder using VAE. The two losses, VAE and the proposed loss are additively combined.

**Strength**

1. Studies goal-conditioned RL, and invariant representation which are important topics in modern RL. The proposed approach is novel to my knowledge.
2. Experimental evidence supports the claim.

**Weakness**

1. There are two strong environment assumptions: the latent dynamics are assumed to be nearly deterministic and the latent state dynamics are independent of the environment. E.g., this task cannot be used for training a navigation bot in a set of houses where the house layout can be different.

2. Weak theoretical support. While prop 1 and prop 2 are helpful, the discussion in lines 148-154 is hand-wavy. A rigorous mathematical argument would have shown that sub-optimality is based on "misalignment" of the decoder and back it up with matching bounds. Also, I am not sure how prop 3 helps. More importantly, the behaviour of the proposed loss $L^{PA}$ is not studied theoretically. Questions such as what is its optimal solution? Does the optimal solution learn the right encoder? Does the empirical minimization achieve this optimal solution? are not addressed.

Some technical questions and concerns are raised below.

**Concerns and Questions**

1. Why is the learned state encoding a vector when a Block MDP has finite states?

2. Why is MMD loss being minimized in $L^{MMD}$ but not in $L^{DIFF}$? What happens if one simply minimizes the L2 distance between encodings in $L^{MMD}$ as in $L^{DIFF}$?

3. What is the exact formalism for near-deterministic dynamics?

4. (Important) If one assumes deterministic dynamics and goals are a subset of state space, then can't one learn open-loop policies which will trivially generalize to all environments? Is the non-trivially coming from things being slightly deterministic? What happens if one continues to use open-loop policies in experiments? Formally, find an open-loop policy that reaches a given goal in the training environment. This should generalize to the test set whenever things are deterministic but also perhaps when things are near-deterministic.

5. Where is "characteristic set of joint distributions " defined? (Line 120)

**Presentation**

1. Consider using condensed notations.  E.g., $x^e(s)$ can simply be written as $x$ whenever the environment or state information is not required which is often the case.

2. Consider adding pseudocode early on. It is easy to get lost since many things are introduced suddenly such as the Skew-Fit algorithm section.

3. Line: 77 "_You_ can regard" seems non-standard. Consider using "One can regard".

4. Line 150, space before the comma in "as the ones in training , it"

5. Line 160 has a letter "s" in $\mathcal{E}_{train}s$ that doesn't parse

6. Line 163: $x_T$ should be replaced by $x^{e}_T$. This is an example where the $e$ can be dropped for the whole episode without any problem.

7. Line 168: Grammar in "stored in a aligned"

**Time Spent Reviewing:**

3

---

> ### Author Response · Authors · 2021-08-10
> **Author Response**
>
> Thank you for your valuable feedback. We address your concerns:
>
> 1. Q: What is the exact formalism for near-deterministic dynamics?
>
> A: We define it as follows: the transition is $\epsilon$-deterministic if $\max_{s, a} H(p(s'|s, a)) \leq \epsilon$ (including $H(p(s_0))$) where $H(\cdot)$ denotes the entropy. Then, by the chain rule of the entropy, we have $\forall s, a, t, H(p(s_t|a_{0:t}) \leq \epsilon (t+1)$. The corresponding illustration of the definition is shown in Figure 2(a).
>
> 2. Q: Why use different losses for $L^{\text{MMD}}$ and $L^{\text{DIFF}}$?
>
> A: When the environment is only near-deterministic ($\epsilon$-deterministic), sample $s^e(a_{0:t})$ may differ from $s^{e'}(a_{0:t})$ owing to stochasity. Consequently, simply minimizing the $L_2$ loss between $z(s^e(a_{0:t}))$ and $z(s^{e'}(a_{0:t}))$ is risky as minimizing it to $0$ may violate the *only if* condition of perfect alignment. Thus, instead, we use the MMD loss on {$z(s^e(a_{0:t})), e \in \mathcal{E}$} samples to match the distributions {$\{\rho(s^e(a_{0:t})), e \in \mathcal{E}\}$} together. This allows the stochasity in the environment.
>
> Besides, $L^{\text{DIFF}}$ is used as a regularizer to prevent the latent representations from collapsing. Thus, we use the $L_2$ loss directly.
>
> 3. Q: Discussion on the $L^{\text{PA}}$ loss.
>
> A: $L^{\text{PA}}$ is composed of two components: $L^{\text{MMD}}$ and $L^{\text{DIFF}}$. $L^{\text{MMD}}$ is the primary component while $L^{\text{DIFF}}$ is an extra regularizer (with much smaller $\alpha_{\text{DIFF}}$ than $\alpha_{\text{MMD}}$). Theoretically speaking,  $L^{\text{MMD}}$ can be minimized to $0$ as a as discussed in answer 2 above. When the environment is nearly deterministic, minimizing $L^{\text{MMD}}$ to $0$ ensures that the encoder almost satisfies the *if* condition, i.e., the same distributions of $\{z(x^e(s)), s \sim \rho(s^e(a_{0:t})\}, e \in \mathcal{E}_{\text{train}}$ and has low entropy. In the mean time, to minimize $L^{\text{DIFF}}$ to a relative large value favors the *only if* condition, i.e., $z(s)$ of two different states are separated. In all, minimizing the $L^{\text{PA}}$ loss can produce a perfect alignment encoder.
>
> In the experiment, we observe that minimizing $L^{\text{PA}}$ achieves the desired encoder. The discussion is listed in section 4.2. As shown in Figure 4, the almost $0$ LER metric over training environments shows that only training with $L^{\text{MMD}}$ on the aligned data can the encoder maps the observations of the same state $s$ to the same latent representation $z$. The $L^{\text{DIFF}}$ also marginally improves the latent representation quality over the training environments and boosts the generalization performance in the testing environments (the two figures in Figure 4 right).
>
>
> 4. Q: Why is the learned state encoding a vector when a Block MDP has finite states?
>
> A: Our algorithm is not limited to finite state problems. Proposition 1 and Proposition 2 also hold in infinite and continuous state space problems. In the experiments, we use a neural network encoder for the latent representation $z$ over the image inputs. Thus, we follow the convention in our description that the latent representation is a vector for consistency.
>
> 5. Q:  Whether open-loop policies will generalize to all environments?
>
> A: No, open-loop policies will not generalize well in practice. The essence of goal-conditioned RL problem is to generalize to unseen goals during testing as training samples are limited. Besides, in practice, goals are represented as high-dimensional inputs. Open-loop policies will fail to generalize during testing because: (1) policies are not trained on unseen goals and thus no previous open-loop policy to use (2) policies can not identify the correct underlying goal state when the environmental factors are changed in testing.
>
> As our theoretical analysis focus on generalization over different environments (different environment factors $b^e$), for simplicity of the model, we omit discussing the policy generalization to unseen goals with finite training samples and we use the expectation over all possibles goals in the formulation.
>
>
> 6. Q: Where is the characteristic set of joint distributions defined?
>
> A: Definition 2 in Appendix A.2.

---

> > ### Comment · Reviewer_S2fL · 2021-09-09
> > **Thanks for your response**
> >
> > Thank you for your response. In particular, clarification on open-loop policy was useful. I am still not sure how prop 3 is useful though. I re-read the paper and have some additional comments for improvements:
> >
> > 1. Please emphasize the formal definition of near-determinism in the paper. I also gather that the start state $s_0$ is deterministic (from equation 1). Please state these assumptions since both of them are quite strong assumptions.
> >
> >
> >
> > 2. The definition of total-variation takes supermum over members of a $\sigma$-algebra. So $A'$ should be a member of $\sigma$-algebra created from $A$ and which needs to be defined.
> >
> >
> >
> > 3. I think the trade-off made in the theory is to cover a more general case (continuous state space, for example) at the cost of rigorous analysis. For example, discussion in Lines 148-154 is hand-wavy, and problems of exploration and finite-sample analysis are ignored. Further, I think section 3.2 needs a theorem statement that states alignment guarantee for $\phi$ in terms of optimizing $L^{PA}$. I would suggest perhaps doing the opposite: solving a more restricted setting with a more complete proof. E.g., one would never really learn a perfectly aligned encoder but only learn an approximately aligned encoder with high probability. A more rigorous analysis should work with the latter notion. Understanding how the approximation errors would cascade would be interesting and helpful. There are algorithms in RL that work asymptotically but fail in practice due to these issues.
> >
> >
> >
> > In summary, the problem being solved is important and the main ideas seem novel, but the lack of complete analysis and rigour leaves a lot to be desired on the theory side of the paper. The presentation needs reworking, in particular, with regards to clearly stating the assumptions and defining terms when they are first referenced (e.g., the term _characteristic set_ is used in line 120 but defined in Appendix). I would so much want to see a complete theory. Overall, I am ambivalent about this one and I don't find enough reasons to further raise my score.

---

> > > ### Author Response · Authors · 2021-09-10
> > > **Thanks for your feedback**
> > >
> > > Thanks for your valuable feedbacks. We will clarify the assumptions (e.g.， near-deterministic dynamic and initial distribution) and remove the tokens' and statements' ambiguities in the next version.
> > >
> > > Furthermore, we will also incorporate a more rigorous analysis on the approximate perfect alignment, which our theoretical framework can be extended to. However, we find that extra mathematical analysis will make the presentation too technical and lengthy. Thus, we choose to state the informal version and analyze the exact perfect alignment in our work. We emphasize our theoretically motivated method, i.e., perfect alignment, that can solve GBMDPs in practical environments. The empirical results (Figure 4) show that the exact perfect alignment over the training environments is indeed acquired.

---

### Decision · Program_Chairs · 2021-09-27

**Decision:**

Accept (Poster)

**Comment:**

The paper focuses on an interesting problem, goal-conditioned policy generalization in block MDPs (a kind of POMDPs where the current state is uniquely identifiable from the current observation, even though a state can emit many different observations). This is a mostly theoretical work, but its experiments are convincing. The reviewers have closely examined the paper's theory and, on the one hand, didn't find errors in it but, on the other hand, found theory gaps in it that could be explained either by the paper being not fully clear or not sufficiently rigorous. Please see the discussions with reviewers WvNf and S2fL, especially reviewer S2fL's "Thanks for your response" comment.

Nonetheless, due to this work being one of the early ones on this topic and being likely to become a stepping stone for further exploration of this area, the metareviewer recommends acceptance, trusting that the authors incorporate the points that came up in the discussion into the final version (it's hard to think of a reason not to do this).